# Dynamic encoding of temperature in the central circadian circuit coordinates physiological activities

Hailiang Li[1,2,3], Zhiyi Li[1,2,3], Xin Yuan[1,2,3], Yue Tian[1,2,3], Wenjing Ye[1,2,3], Pengyu Zeng[1,2,3], Xiao-Ming Li[2,3,4] & Fang Guo[1,2,3] ✉

The circadian clock regulates animal physiological activities. How temperature reorganizes circadian-dependent physiological activities remains elusive. Here, using in-vivo two-photon imaging with the temperature control device, we investigated the response of the *Drosophila* central circadian circuit to temperature variation and identified that DN1as serves as the most sensitive temperature-sensing neurons. The circadian clock gate DN1a's diurnal temperature response. Trans-synaptic tracing, connectome analysis, and functional imaging data reveal that DN1as bidirectionally targets two circadian neuronal subsets: activity-related E cells and sleep-promoting DN3s. Specifically, behavioral data demonstrate that the DN1a-E cell circuit modulates the evening locomotion peak in response to cold temperature, while the DN1a-DN3 circuit controls the warm temperature-induced nocturnal sleep reduction. Our findings systematically and comprehensively illustrate how the central circadian circuit dynamically integrates temperature and light signals to effectively coordinate wakefulness and sleep at different times of the day, shedding light on the conserved neural mechanisms underlying temperature-regulated circadian physiology in animals.

The circadian clock orchestrates animals' physiological activities, including behavior and metabolism, in response to environmental fluctuations. This conservative mechanism enables animals to regulate their physiological functions in a well-organized manner at different times by perceiving variations in environmental temperature and light[1–3]. By synchronizing their internal clocks with external cues, such as light and temperature, animals can optimize energy conservation, adapt to environmental changes, and gain evolutionary advantages[4,5].

Understanding how central circadian neurons perceive and respond to environmental temperature changes, particularly within the suprachiasmatic nucleus (SCN) of mammals, remains a formidable challenge. However, *Drosophila* provides an advantageous system for studying temperature-related influences due to its 150 circadian neurons expressing four core clock genes (*Clock, Cycle, period*, and *timeless*), and its shared circadian activity with humans[6–10]. Notably, fruit flies, unlike homeothermic animals, exhibit heightened sensitivity to temperature fluctuations, resulting in adjustments in sleep-wake patterns[11,12]. Previous studies have revealed the significant impact of elevated temperatures on diurnal sleep-wake patterns, including prolonged siesta duration, delayed onset and offset of the evening peak (E peak), and reduced nighttime sleep[13–15]. Conversely, decreased temperatures augment daytime locomotion and facilitate nocturnal sleep in *Drosophila*[16].

[1]Department of Neurobiology, Department of Neurology of Sir Run Run Shaw Hospital and School of Brain Science and Brain Medicine, Zhejiang University School of Medicine, Hangzhou 310058, China. [2]MOE Frontier Science Center for Brain Research and Brain-Machine Integration, State Key Laboratory of Brain-machine Intelligence, Zhejiang University, 1369 West Wenyi Road, Hangzhou 311121, China. [3]NHC and CAMS Key Laboratory of Medical Neurobiology, Zhejiang University, Hangzhou 310058, China. [4]Department of Neurobiology and Department of Psychiatry of the Second Affiliated Hospital, Zhejiang University School of Medicine, Hangzhou 310058, China. ✉e-mail: gfang@zju.edu.cn

Recent research has focused primarily on how temperature affects DN1s[7,13,17], which are hierarchically organized neurons that lie downstream of lateral clock neurons and serve as intermediaries to transmit circadian information from the pacemaker to downstream brain regions that control behaviors[18]. DN1a and DN1p are two subgroups of DN1s that play critical roles in regulating behaviors in response to temperature changes[17]. DN1as are located on the dorsal-anterior side of the fruit fly brain and can be suppressed by low-temperature information transmitted from the upstream GABAergic TPN-IIs[16]. On the other hand, DN1ps can be activated in high temperature transmitted from TrpA1[+] ACs to modulate nighttime sleep[13,17]. In addition, research has shown that DN1ps mediate the prolonged morning wakefulness induced by high temperature[17]. Meanwhile, LPN, as another downstream target of ACs, regulates the high temperature-evoked delay of E peak onset[16]. Previous studies have primarily examined the input of temperature-transmitting neurons, such as TPNs and ACs, on DN1as and DN1ps. However, those studies did not offer a comprehensive perspective of how temperature fluctuations are integrated into the entire circadian circuit, including DN1a, DN1p, and the downstream pacemakers, to regulate diverse behavioral outputs at distinct circadian timepoints[7,16].

To investigate circadian neurons' response to temperature changes, we developed a precise temperature control device and metal flyplate. Our findings emphasize DN1as as highly temperature-sensitive neurons in the fly brain, with activation at high temperatures and inhibition at low temperatures. Notably, DN1a's temperature response exhibits circadian variation, being weaker during the day and progressively heightened at night, linked to levels of the circadian protein PER. Calcium imaging experiments demonstrated synchronized DN1as' calcium oscillations with temperature fluctuations, reinforcing their role as central circadian temperature sensors. We elucidated a circadian neuronal circuit involving DN1as, LNds, and DN3s that modulates temperature-dependent physiological activities. DN1a acts as a hub for rhythmic temperature integration, regulating locomotion and sucrose intake in response to temperature changes. These findings enhance our understanding of how circadian neuronal circuits integrate temperature information to finely regulate daily physiological activities.

## Results

### A screen for temperature-sensitive circadian neurons in *Drosophila*

In order to systematically investigate the response of circadian neurons (Fig. 1a) in live flies to temperature fluctuations, we developed a two-photon imaging system that could monitor real-time calcium changes in these neurons, in conjunction with a precise temperature control device (Fig. 1b). To achieve this, we designed a 3D-printed metal flyplate that was directly connected to a Peltier module, allowing for simultaneous control of both saline and ambient temperature (Supplementary Fig. 1a). Micro-surgery was performed on the fly cuticle to expose the circadian neurons, which were labeled with GCaMP6s[19] driven by *Clk856-GAL4*[20], enabling us to visualize most of the dorsal circadian neurons, including DN1ps, DN2s, DN1as, and a part of DN3s. Our results showed that cooling from 24 to 18 °C significantly reduced calcium levels in DN1as, while other circadian neurons did not display noticeable changes (Fig. 1c−e, Supplementary Video. 1, Supplementary Fig. 1h). On the other hand, heating from 22 to 30 °C caused a more obvious calcium increase in DN1as than in other subsets of circadian neurons (Fig. 1f−h, Supplementary Video. 2, Supplementary Fig. 1i−l). These findings suggest that DN1as are the most sensitive circadian neurons to changes in ambient temperature.

To characterize DN1a responses to different temperature changes in detail, we generated a split GAL4 (*R14F03-GAL4.AD ∩ VT002963-GAL4.DBD*) line specifically labeling DN1as (Fig. 1i, Supplementary Fig. 1b−d). Our findings indicate that at specific ZT times, cold

temperature resulted in a reduction of calcium levels in DN1as, while warm temperature increased calcium levels. A control experiment with the calcium-insensitive fluorescent protein GFP expressed in the same DN1as showed no changes in fluorescence intensity in response to temperature fluctuations (Supplementary Fig. 1e−g). These observations suggested that DN1as are continuously sensing and monitoring the ambient temperature, as they are inhibited by cold stimuli and activated by hot stimuli.

To explore the temperature response range of DN1as, we exposed the flies to various temperatures and monitored the responses of DN1as. We found that DN1as were suppressed by a 2 °C-cooling step and inhibited to saturation at temperatures 6 °C below the 24 °C baselines (Fig. 1j, k). Moreover, we observed that DN1as could be activated by a 2 °C-heating step and driven to saturation during an 8 °C-heating step above the 22 °C baselines (Fig. 1j, k). In summary, our results provide substantial evidence that DN1as are adept at sensing environmental temperature fluctuations. The evidence underscores the potential importance of DN1as in mediating the fly's circadian response to changing temperature conditions, thereby facilitating appropriate adjustments in their physiological activities.

### Evaluation of DN1a diurnal calcium response to temperature variations

The natural environment exhibits a diurnal temperature fluctuation pattern, with higher temperature occurring at day and lower temperature at night. It was of interest to determine whether DN1as, the most temperature-sensitive circadian neuron in the Drosophila brain, can be entrained by the diurnal variation of ambient temperature and exhibit a rhythmic activity pattern[21]. To achieve this, we employed a long-term in vivo calcium recording method in live flies, allowing us to monitor real-time DN1a calcium dynamics. This involved tethering flies with a cuticle window exposing the DN1as for two-photon microscopy, followed by the quantification of average calcium levels in the DN1as every 30 min (Supplementary Fig. 2a). Our findings unveil a diurnal pattern in the calcium levels of DN1as, characterized by an increase during the day followed by a decrease at night (Fig. 2a, b). This rhythmic fluctuation of DN1a calcium levels aligns with the natural light-dark cycle and suggests potential regulation by an intrinsic circadian clock.

Since DN1as display varying neuron activity during the LD cycle, we wanted to determine if they respond differently to heating or cooling cycles at different circadian times. Indeed, we observed that DN1as showed repeated activation or inhibition to heating or cooling cycles throughout the middle night (ZT17-19). In contrast, DN1as displayed relatively little response to temperature fluctuations during the middle day (ZT6-8) (Supplementary Fig. 2b, c). We further examined DN1a's response to temperature changes at more ZT times and found that DN1as exhibited distinct circadian-dependent responses to temperature changes, with weak responses observed during the early morning (ZT0-2) and afternoon (ZT6-8), and strong responses during the evening and midnight (ZT10-14 and ZT17-19) (Fig. 2c, d). Intriguingly, when monitoring the calcium dynamics and fly locomotion on the air-supported ball at the same time, we revealed that cold-induced DN1a inhibition at night was associated with reduced fly motility (Supplementary Video. 1). On the other hand, the hot temperature at night increased both DN1a activity and fly motility (Supplementary Video. 2). Taken together, our findings suggested that temperature can impact the calcium activity of circadian neurons, thereby influencing an animal's sleep-wake level.

The diurnal variation in DN1a's responses to temperature changes may be due to alteration in inhibitory input signals from upstream GABAergic TPN-IIs, or the influence of the endogenous circadian clock on DN1a excitability[16]. To determine whether TPN-IIs exhibit a similar diurnal variation in their responses to temperature changes, we examined the calcium levels at TPN-II axon terminals. Consistent with

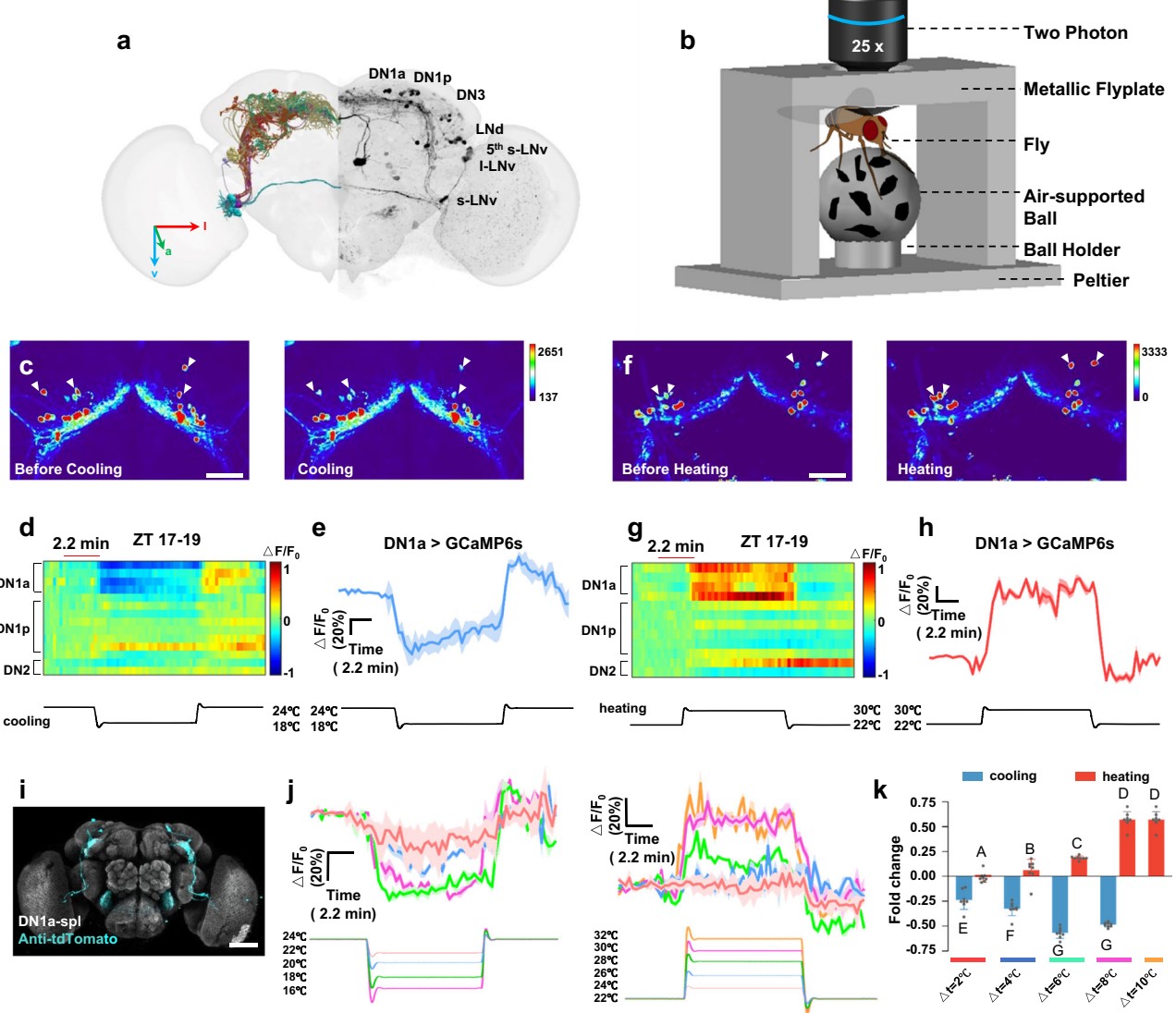

**Fig. 1 | *Drosophila* circadian neurons respond to changes in ambient temperature. a** Schematic of the *Drosophila* circadian neuronal circuit. Left panel is the EM reconstruction of circadian neurons and right panel is a *Clk856-GAL4>GCaMP6s* brain stained with GFP antibodies. **b** Schematic of the two-photon imaging system with precise temperature control and fly-on-ball setups. **c**, **f** Representative pseudocolor images of calcium responses of *Clk856-GAL4>UAS-GCaMP6s* flies before and after cooling (**c**) and heating (**f**). white arrow indicates DN1as. **d**, **g** Heat map of the temporal calcium activities of circadian neurons to cooling (24–18 °C, **d**) and heating (22–30 °C, **g**). The temperature change was labeled under the panel. **e**, **h** The representative GCaMP traces ($\Delta F/F_0 \pm$ SEM) of DN1as in response to cooling (blue, $n = 16$, **e**) and heating (red, $n = 16$, **h**). **e** related to (**d**). **h** related to (**g**).

**i** Expression pattern of the DN1as driven by *DN1a-spl* in the brain revealed by anti-tdTomato (cyan) and anti-NC82 (gray). Scale bar, 20 μm. **j** The representative GCaMP traces ($\Delta F/F \pm$ SEM) of DN1as in response to cooling (left) and heating (right) with steps of different amplitude (red: $\Delta T = 2$ °C; blue: $\Delta T = 4$ °C; green: $\Delta T = 6$ °C; purple: $\Delta T = 8$ °C; orange: $\Delta T = 10$ °C, $n = 8$). **k** Quantification of the relative fold change of calcium activities of DN1as in (**j**). Data are presented as mean in $\Delta F/F_0 \pm$ SEM in the histogram; Statistical analysis was conducted using One-Way ANOVA followed by Tukey post-test for multiple comparisons. The letters A, B, C, D, E, F and G above the histograms denote significantly different means within each of the two groups, $p < 0.05$. Specific $p$ values corresponding to this figure are reported in the Source Data. $n = 8$.

previous studies, we observed that the cold condition increased the calcium levels of TPN-II axons. However, we did not observe a significant diurnal difference in TPN-II temperature sensitivities (Supplementary Fig. 2d–f).

We then evaluated the contribution of the endogenous circadian clock to the DN1a temperature responses. The circadian proteins PER and TIM periodically undergo degradation during the daytime and accumulate at night, which may affect the daytime dead zone and nocturnal sensitive window of DN1a temperature responses. Our results showed that *per* mutant flies exhibited abolished diurnal variation of DN1a temperature responses and were phase-locked to a daytime-like modest response observed in wild-type flies. This

response was characterized by a rapid decline in DN1a calcium levels and a quick return to baseline in response to cold temperature (Fig. 2e–g, Supplementary Fig. 2g–i). This data supported our hypothesis that the circadian clock gated the temperature response, as *per* mutant flies mimic the daytime conditions in which wild-type flies have low PER levels in circadian neurons[22]. Surprisingly, rescuing the PER expression cell autonomously in only DN1as was sufficient to restore the diurnal variation of the DN1a temperature response. Notably, the GAL4-driven PER rescue led to an overexpression of PER levels in DN1as (Supplementary Fig. 2g–i), resulting in a greater cold-induced DN1a inhibition during the daytime (Fig. 2e–g, Supplementary Fig. 2j–l). In addition, a strong negative correlation between baseline

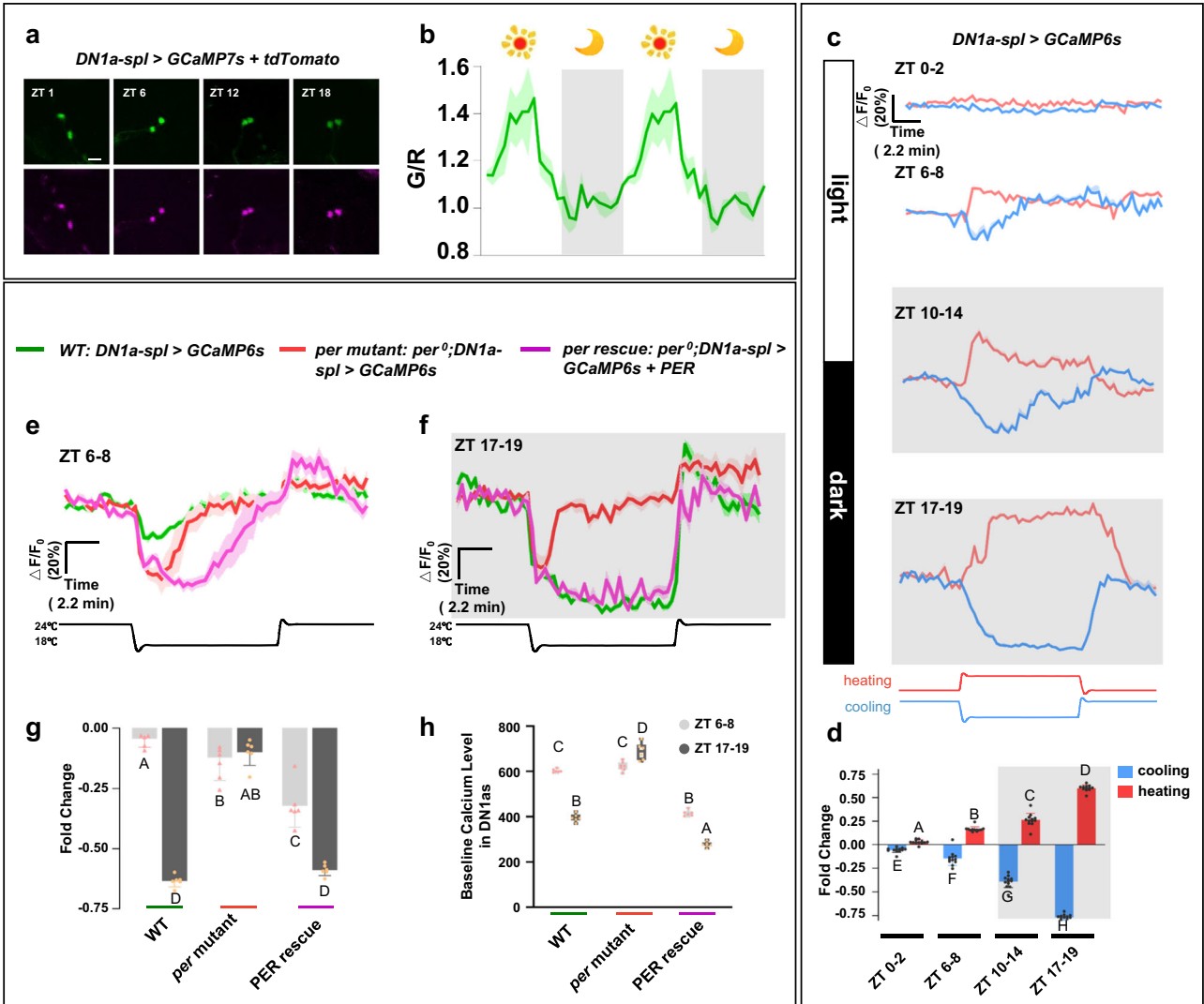

**Fig. 2 | Characterization of DN1a diurnal calcium response to ambient temperature variations. a** Representative images of GCaMP (top panels) and tdTomato (bottom panels) fluorescence intensity in the cell bodies of DN1as at four circadian times. $n = 6$. Scale bars, 10 μm. ZT time points were labeled in panels. **b** The averaged calcium oscillation pattern of DN1as through long-term in vivo calcium recording in live *DN1a-spl>GCaMP6 + tdTomato* flies. The data were double-plotted to better visualize the pattern. Dark shades indicate lights off (night). $n = 6$. Data are presented as mean in $\Delta F/F_0 \pm$ SEM in the line graphs. **c** The representative GCaMP traces ($\Delta F/F_0 \pm$ SEM) of DN1as in response to cooling (blue, $n = 12$) and heating (red, $n = 12$) at 4 different times. Dark shades indicate lights off (night). **d** Quantification of relative fold change of calcium activities of DN1as in (**c**). **e, f** The representative GCaMP traces ($\Delta F/F_0 \pm$ SEM) of DN1as in response to cooling in flies at ZT 6–8 (**e**) and ZT 17-19 (**f**). green curve: wild type; red curve: *per* mutant; purple curve: PER rescue. Dark shades indicate lights off (night); $n = 6$. **g** Quantification of the relative fold change of calcium activities of DN1as in (**e, f**); $n = 6$. **h** Basal GCaMP6s signals in DN1as at two circadian time points in (**e, f**); $n = 6$. **d, g, h** For all histograms, data are presented as mean in $\Delta F/F_0 \pm$ SEM in the histogram; Statistical analysis was conducted using One-Way ANOVA followed by Tukey post-test for multiple comparisons. The letters A, B, C and D above the histograms denote significantly different means within each of the two groups, $p < 0.05$. Specific $p$ values corresponding to this figure are reported in the Source Data.

DN1a calcium activity and PER levels was observed, where *per* mutant flies consistently displayed elevated calcium levels during both day and night, while DN1a PER rescue/overexpression flies exhibited a significantly low calcium level during both day and night (Fig. 2h).

To gain deeper insight into the mechanism by which the circadian clock regulates calcium levels in DN1as, we performed an RNAi screen targeting ion channel proteins potentially involved in circadian regulation. We specifically examined proteins such as SERCA (sarco/endoplasmic reticulum Ca$^{2+}$-ATPase)[23], Shaker[24], Nocte[25], and Narrow abdomen[26] (Supplementary Figs. 3, 4). Our study showed that specific knockdown of SERCA in DN1as results in a loss of circadian variation in calcium levels, with significantly lower basal calcium levels both day and night (Supplementary Figs. 3d, 4e). This finding is consistent with a previous study showing that SERCA RNAi in circadian neurons reduces the amplitude of calcium fluctuations[23]. In addition, these flies showed

a greater response to temperature changes both day and night (Supplementary Figs. 3a–c, 4c, d). Meanwhile, under LL conditions, the DN1a SERCA RNAi flies show undetected PER levels but still manifest a SERCA knockdown-like phenotype with consistently lower basal calcium levels and enhanced cold-induced calcium reduction throughout the circadian cycle (Supplementary Fig. 4a, b). This indicates that SERCA is epistatic to PER as a crucial molecular effector influencing DN1a calcium levels. These results suggest that the circadian clock may modulate calcium levels in DN1as through SERCA-related endoplasmic reticulum calcium stores, thereby influencing the neurons' circadian response to temperature (Supplementary Fig. 4f). Taken together, our findings elucidate the mechanism by which DN1as integrates circadian clock and temperature information, enabling dynamic processing and transmission of this signal to the central circadian pacemakers.

## Morphological and functional characterization of DN1a-LNds and DN1a-DN3 circuits

DN1as are well-suited for integrating persistent diurnal temperature changes into the circadian circuitry, as they are targeted by slow-adapting temperature sensory projection neurons rather than transient fluctuations[16]. Previous anatomical studies have shown that the DN1as in the dorsomedial protocerebrum send fibers to the accessory medulla to innervate central pacemakers such as the LNvs[21] (Supplementary Fig. 5a). To determine the primary downstream targets of DN1as, we utilized the neuPrint analysis tool for EM connectomics in the female *Drosophila* brain and found that DN1as primarily send synaptic output to other circadian neurons such as DN3s and E-cells including the 5th s-LNvs and LNds[27] (Fig. 3b). Notably, the KCs, which form part of the α'β' lobe of Mushroom Bodies, are among the non-circadian downstream targets of DN1as (Fig. 3b). To validate this

connection in male flies, we employed the trans-synaptic tracing technique *trans*-Tango[28]. Our data revealed that the primary post-synaptic targets of DN1as appear to be the neurons DN3, LNds, LNvs, and KCs (Fig. 3c). Further, PER staining confirmed that these DN1a post-synaptic targets are indeed circadian neurons (Figs. 3d, 4a).

The DN1a-E cell circuit is an intriguing focus of study because the LNds and 5th s-LNv are the primary neurons innervated by DN1as, and these E-cells control the evening locomotion peak, which is dynamically reshaped by temperature[5,14,15]. In addition, morphological data suggested a robust connection between DN1a axons and LNd dendrites (Fig. 3e). To gain a better understanding of the functional connection of this circuit, we analyzed the neurotransmitter released by DN1as. Our results, in agreement with previous studies, indicate that DN1as are glutamatergic neurons[29] (Fig. 3a). Given that inhibitory glutamate receptors such as mGluRA and GluClα are expressed in

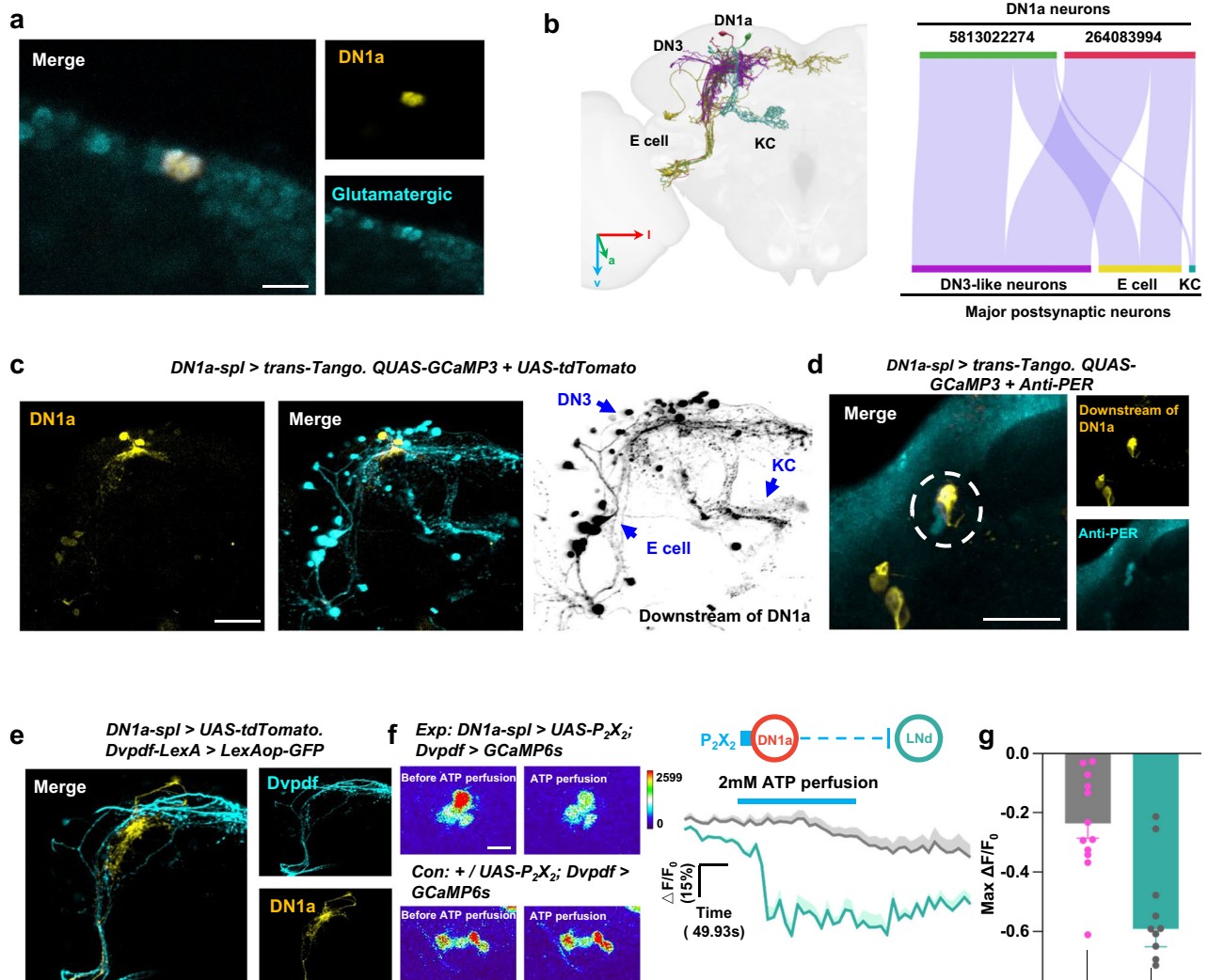

**Fig. 3 | Glutamatergic DN1as primarily send inhibitory projection to E cells.**
**a** DN1as are glutamatergic neurons. Anti-RFP (glutamate) and anti-GFP (DN1a) staining were visualized in the brain of flies. Scale bars, 20 μm. **b** Schematic of the major downstream neurons of DN1as reconstructed in fly brain template. The left panel is the EM reconstruction of DN1as and downstream neurons. The right panel displays a Sankey histogram depicting the synaptic connections between DN1as and major postsynaptic neurons, as determined by the Hemibrain connectome dataset. a: anterior. l: latter. v- ventrolateral. **c** Identifying postsynaptic connections of DN1as using *trans*-Tango. From left to right: Presynaptic DN1as (yellow), merged images, and postsynaptic *trans*-Tango signal (cyan / black). The genotype was labeled above the

panel. Scale bars, 20 μm. **d** Representative PER and GFP double staining of *DN1a-spl > trans*-Tango brain. Anti-PER (cyan) and anti-GFP (LNd: yellow) staining were visualized in the brain of flies. White circle indicates LNds. Scale bar, 50 μm. **e** Double staining of GFP (cyan) and RFP (yellow) in *Dvpdf-LexA > LexAop-GFP, DN1a-spl>UAS-tdTomato* brain. Scale bars, 50 μm. **f** Representative pseudocolored images of calcium responses of LNds before and after ATP perfusion activation of DN1as (left); Scale bars, 10 μm. Representative GCaMP traces ($\Delta F/F \pm$ SEM) of LNds in response to DN1as activation (right, $n = 12$). Con: grey; Exp: green. **g** Quantification of relative max calcium activities of LNds in (**f**). Data are presented as mean in Max$\Delta F/F_0 \pm$ SEM in the histogram; 2-tailed *t*-test; ****$p < 0.0001$, $p = 0.000145$. $n = 12$.

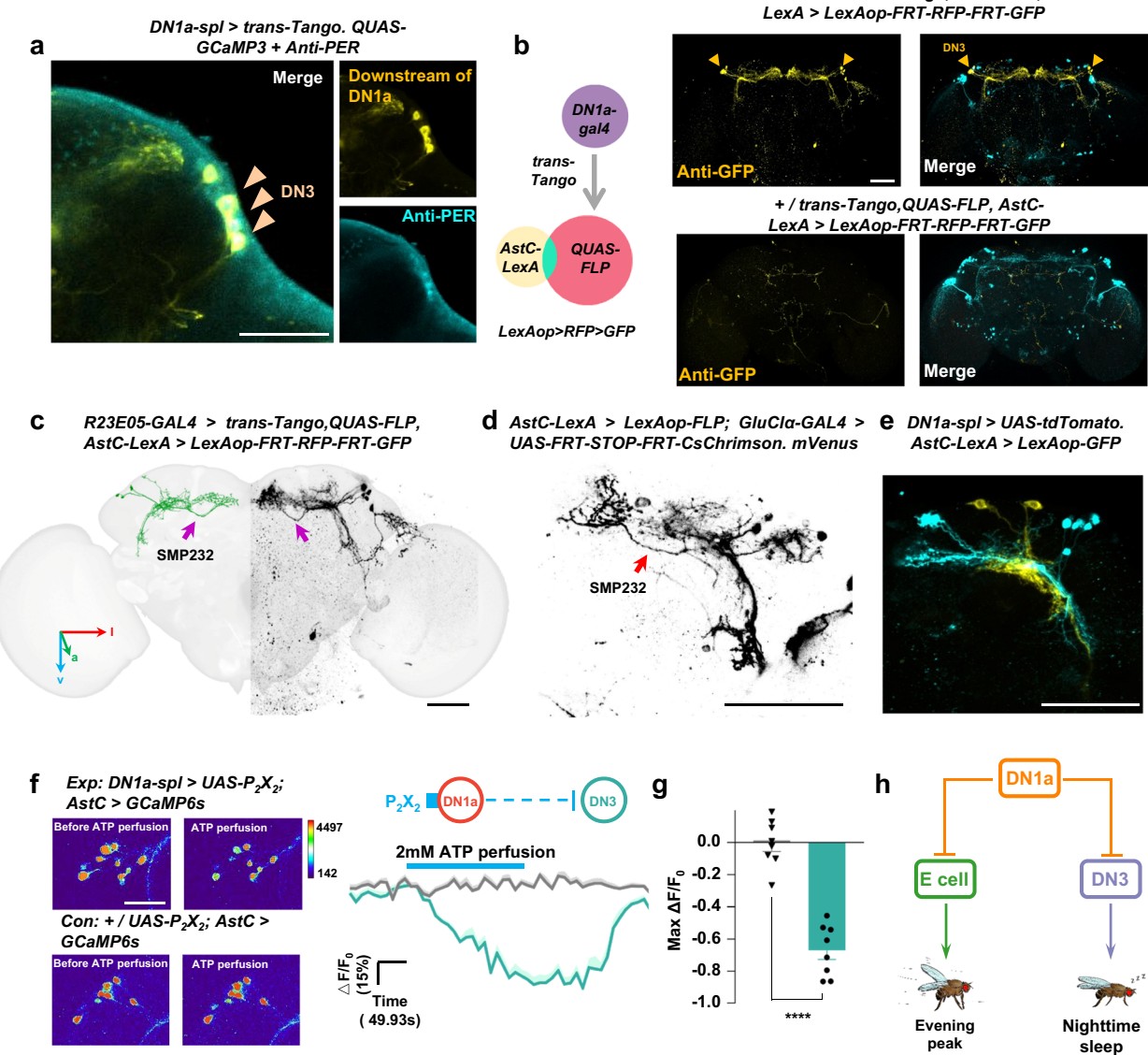

**Fig. 4 | DN1as inhibits a subgroup of DN3s including SMP232. a** Representative PER and GFP double staining of *DN1a-spl > trans-Tango* brain. Anti-PER (cyan) and anti-GFP (DN3: yellow) staining were visualized in the brain of flies. Yellow arrows indicate DN3s. Scale bar, 20 µm. **b** Direct synaptic contacts between DN1s and DN3s revealed by restricted *trans*-Tango. The experimental and control brains were stained with anti-GFP (yellow) and anti-RFP (cyan), respectively, to identify that the DN3s were postsynaptic to DN1as. Anti-GFP-labeled neurons are downstream of DN1as (upper). Anti-RFP-labeled neurons are non-downstream neurons (lower). The yellow arrow indicates DN3s; Scale bars, 50 µm. **c** Anatomic characterization of the SMP232-like DN3s postsynaptic to DN1as revealed by restricted *trans*-Tango in (**b**). The left panel is the EM reconstruction of DN3s and the right panel is a GFP staining of downstream of DN1a by restricted *trans*-Tango. Purple arrow indicates

SMP232-like DN3s; Scale bar, 50 µm. **d** SMP232-like DN3s are labeled by intersection between *GluClα-GAL4* and *AstC-LexA*. The red arrow indicates SMP232-like DN3s; Scale bar, 20 µm. **e** Double staining of GFP (cyan) and RFP (yellow) in *AstC-LexA > LexAop-GFP, DN1a-spl > UAS-tdTomato* brain. Scale bar, 20 µm. **f** Representative pseudocolored images of calcium responses of DN3s before and after ATP perfusion activation of DN1as (left); Scale bars, 20 µm. Representative GCaMP traces ($\Delta F/F$) of DN3s in response to DN1as activation (right, $n = 8$). Con: grey; Exp: green. **g** Quantification of relative max calcium activities of DN3s in (**f**). Data are presented as mean in Max$\Delta F/F_0 \pm$ SEM in the histogram; two-tailed *t*-test; ****$p < 0.0001$, $p = 4.95044E-07$. **h** Model of two inhibitory DN1as-DN3s and DN1as-LNds circuits based on our functional imaging data.

E-cells[29], we postulated that DN1as may transmit temperature and circadian signals to inhibit E-cells. To test this hypothesis, we engineered flies expressing the ATP-gated cation channel $P_2X_2$ in DN1as and GCaMP6s in LNds. Upon activation of DN1as via ATP perfusion, a marked decrease in calcium levels in LNds was observed (Fig. 3f, g). This observation supports the presence of a potent inhibition from DN1as to LNds.

The results of both the Neuprint analysis and our *trans*-Tango experiment provided robust evidence for a connection between DN1a and DN3s (Fig. 4a). The DN3s have been previously shown to exhibit

maximum calcium levels during the night and serve as sleep-promoting neurons[30]. Thus, we propose that DN1as may modulate sleep patterns in response to nocturnal temperature changes through its interaction with DN3s. To better understand the specific DN3s that are postsynaptic to DN1a, we utilized our restricted *trans*-Tango technique, which allowed us to identify 7-8 allatostatin-C positive (AstC+) DN3s per hemisphere projecting to the dorsomedial protocerebrum and the superomedial protocerebrum (SMP) region (Fig. 4b–d, Supplementary Fig. 5b, Supplementary Video. 3). Morphological co-staining further revealed that DN1a branches are

intertwined with DN3 fibers (Fig. 4e). Additionally, by activating DN1a-expressing P$_2$X$_2$ receptors with ATP perfusion, we observed the inhibition of subsets of DN3s (Fig. 4f, g). This suggests that the DN1a-DN3 formed an inhibitory circuit (Fig. 4h).

Furthermore, we generated an AstC$^+$ DN3 split-GAL4 driver (*R67F03-GAL4.AD∩VT002670-GAL4.DBD*) (Supplementary Fig. 5c–e), and our staining results revealed strong morphological connectivity between DN3 and LNd fibers (Supplementary Fig. 5f). Previous research has shown that AstC peptide released from DN3s can inhibit LNds[31], which is consistent with our findings demonstrating that activating DN3s with ATP perfusion can suppress LNds (Supplementary Fig. 5g, h). Taken together, our results showed that the inhibitory DN1as are linked to a bidirectional circuit: the activity-promoting LNds

and the sleep-promoting DN3s, and the switch of rerouting may depend on circadian time to reshape sleep-wake patterns in response to temperature fluctuations (Supplementary Fig. 5i).

## The DN1a-LNd circuit regulates cold-induced E peak onset advance

Building upon the observation that LNds serve as the primary post-synaptic neurons for temperature-sensitive DN1as (Fig. 5a), our objective was to investigate their direct sensitivity to temperature fluctuations. Previous research has established that LNd calcium levels exhibit peaks during the night[23]. Furthermore, our GRASP data reveal that the synaptic connections between DN1a and LNd are plastic, becoming significantly stronger during nighttime. This combination of

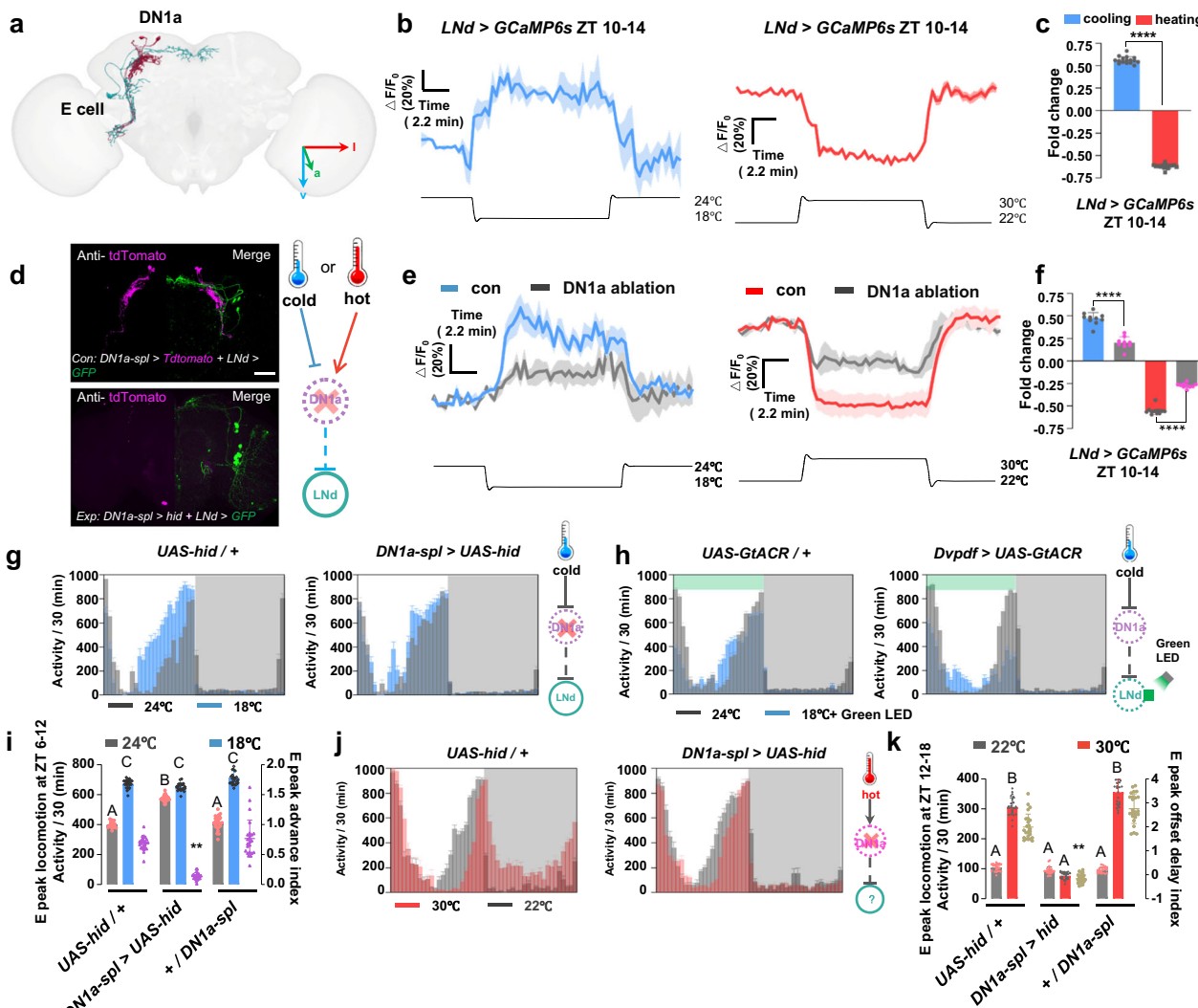

**Fig. 5 | DN1as transmit temperature signals to E cells and modulate the timing of E peak. a** Examples of DN1as and E cells obtained from electron microscopy (EM) in neuPrint database. **b** Representative GCaMP traces ($\Delta F/F_0 \pm$ SEM) of LNds in response to cooling (left, blue, $n = 16$) and heating (right, red, $n = 16$). **c, f** Quantification of the relative fold change of calcium activities of LNds in (**b**, **e**). Data are presented as mean in $\Delta F/F_0 \pm$ SEM in the histogram; two-tailed *t*-test; ****$p < 0.0001$ (**c**, p = 2.03503E-28; **f**, left panel: $p = 1.00641$E-07, right panel: $p = 4.11048$E-14). **d** Representative double staining images of Anti-GFP and Anti-tdTomato in fly brains of control group (top panel) and experimental group (bottom panel). Brains were stained with anti-GFP (green) and anti-tdTomato (magenta); Scale bars, 50 µm. **e** representative GCaMP traces ($\Delta F/F_0 \pm$ SEM) of LNds in response to cooling (left panel) and heating (right panel) in the intact fly (blue, $n = 9$) and DN1a silenced fly (gray, $n = 10$). **g, h, j** Representative locomotor activity

profiles of control ($n = 23$) and DN1a-ablated (**g**, $n = 24$) and LNd-inhibited (**h**, $n = 24$) flies at 24 °C (gray column), 18 °C (blue column) and 30 °C (**j**, red column, $n = 24$). Activity summed in 30-min bins were plotted. Bar graphs are averages ± SEM; Dark shades indicate lights off (night). **i, k** Quantification of E peak locomotion and E peak advance index (magenta) and E peak offset delay index (brown) in (**g, j**). The gray, blue and red column indicates flies' locomotion at 22 °C or 24 °C,18 °C and 30 °C; the purple and brown plots indicate E peak advance/ offset delay index; Data are presented as mean in $\Delta F/F_0 \pm$ SEM in the histogram; Statistical analysis was conducted using One-Way ANOVA followed by Tukey post-test for multiple comparisons. The letters A, B and C above the histograms denote significantly different means within each of the two groups, $p < 0.05$. Specific $p$ values corresponding to this figure are reported in the Source Data. **i** related to (**g**). **k** related to (**j**).

findings underscores the intricate relationship between DN1as and LNds in responding to temperature changes.

Based on the finding that LNds are the primary post-synaptic neurons of temperature-sensitive DN1as (Fig. 5a), we aimed to explore their direct sensitivity to temperature fluctuations. Prior research has established that LNd calcium levels display peaks during the night[23]. Additionally, our GRASP data reveal plasticity in the synaptic connections between DN1a and LNd, with these connections significantly strengthening around the evening (Supplementary Fig. 6a, b). Temperature changes may potentially affect the regulation of the onset and offset of the nocturnal locomotor peak under LNd control[5,14,15]. Therefore, we specifically chose the time window of ZT10-14 to monitor the calcium response of LNds to temperature fluctuations.

Consistent with our observation that stimulation of DN1as with ATP-gated $P_2X_2$ channels[32] can inhibit LNds, heating, which also activates DN1as, reduces calcium levels in LNds. Conversely, cooling, which effectively inhibits DN1as, significantly enhances LNd activities (Fig. 5b, c). To investigate whether DN1as are the primary temperature input neurons regulating LNds, we genetically ablated DN1as and evaluated the LNd response to temperature changes. We also ensured that the DN1a ablation did not affect the morphology of the LNds (Fig. 5d). Results showed that DN1a ablation, as well as blockage of DN1a glutamate release, significantly reduced the LNd calcium response to both cold and hot temperature (Fig. 5e, f, Supplementary Fig. 6c, d, 9a–d). However, the remaining LNd response to temperature changes suggests that other temperature input sources, such as LPN, may still modulate LNd activity even in the absence of DN1as.

The next question that arises is how the DN1a-LNd circuit processes temperature information to regulate locomotor behavior. To address this, we utilized a customized Flybox that incorporated a precise temperature control system (Supplementary Fig. 7a). The system was achieved by attaching a 3D-printed metal 96-well plate to a Peltier device, which allowed us to fast and accurately control the ambient temperature around the flies in the wells while maintaining optogenetic and video recording capabilities (Supplementary Fig. 7b–e).

When cooled from 24 °C to 18 °C, control flies exhibited an advanced E peak and a reduction in afternoon sleep as indicated by previous studies[14,16]. This phenomenon may be explained by cold lowering DN1a calcium levels, releasing LNds from DN1a inhibition, and boosting evening locomotion. To test this hypothesis, we genetically ablated DN1as or blocked DN1a release. We expected that under normothermic conditions, these manipulations would replicate the effects of cold temperature and result in an earlier onset of E peak. The results showed that indeed, DN1a-silenced (Fig. 5g, i, Supplementary Fig. 7f, h) or DN1a output-blocked (Supplementary Fig. 7k, l) flies exhibited an advanced E peak onset even at 24 °C, similar to the responses observed in the control group under cold conditions.

In addition, we found that the cold temperature no longer induced a further advanced E peak in the DN1a-silenced flies than in the control group, which may be due to DN1a ablation blocking the transfer of cold information. RNAi-mediated knockdown of the DN1a transmitter glutamate also blocked both cold-induced LNd activation and cold-induced E peak advancement (Supplementary Fig. 9e, f). Interestingly, acute optogenetic inactivation of LNds via *GtACR*[33] completely suppressed the E peak at low temperatures under green light exposure (Fig. 5h, Supplementary Fig. 7g, j, i). Additionally, chronic silencing of LNds using *Kir2.1* led to the suppression of the E peak advance at low temperatures (Supplementary Fig. 7m–o). The findings suggest that cold causes the E peak to advance through LNds activation. Together, our data collectively suggest that the inhibitory DN1a-LNd circuit integrates temperature information and modulates cold-induced evening locomotion in flies.

To determine the role of DN1a in shaping the diurnal sleep/wake pattern in response to high temperature, we genetically silenced DN1a

using ablation or output blockade. Surprisingly and importantly, we found that silencing DN1a (Fig. 5j, k, Supplementary Fig. 8a–c) or blocking DN1a output (Supplementary Fig. 8e, f, 8g, h) entirely prevented the heat-induced delay in E peak offset after ZT12 as well as the heat-induced reduction in nocturnal sleep when heated from 22 °C to 30 °C (Supplementary Fig. 8b). These results suggested that the DN1a-silenced flies do not respond to high temperature during the night.

However, the delayed onset of the E peak caused by high temperature remained in DN1a-silenced flies, which may be due to the role of LPN in transmitting hot temperature information in the absence of DN1as[15]. Despite this, the E peak in the DN1a-silenced groups was considerably broader than that in the control group (Supplementary Fig. 8d), supporting our findings that the LNds in DN1a-silenced flies were not as effectively inhibited by hot temperature as the controls. In addition, the DN1a-LNd circuit cannot account for the reduction in nocturnal sleep induced by high temperature because LNd activity is low at night[23], and ablation of DN1as should promote LNd activation and increase locomotion rather than sleep (Fig. 5g, i, Supplementary Fig. 7f, h). On the other hand, prior studies have demonstrated that DN3 activity starts to rise during the night and is involved in a recurring sleep-promoting mechanism[23,30]. We therefore hypothesize that the altered DN1a-DN3 circuit activity may be the root of the heat-induced reduction in nocturnal sleep.

## Darkness switch on the DN1a-DN3 circuit to regulate heat-induced E peak offset delay and nocturnal sleep reduction

EM connectome analysis revealed that DN3-like neurons (SLP266, SLP267, and SMP232, Supplementary Fig. 10a) are the largest downstream group of DN1as (Figs. 3b, 6a). Additionally, the data indicated that DN1as and light input neurons, such as aMe and s-LNv, potentially exert a significant impact on DN3-like neurons, which implies a probable correlation between DN3 activity and its modulation by light and temperature[14,21,34]. To explore the function of the DN1a-DN3 circuit in shaping the sleep-wake profile at high temperature, the calcium activity of DN3s was recorded in response to temperature changes using the ZT17-19 night window, as we observed a marked nocturnal sleep reduction at high temperature. Our results suggested that DN3s exhibit an opposite temperature-induced response to DN1as, where cooling can activate DN3s while heating significantly inhibits DN3s (Fig. 6b, c). Although DN1a ablation did not affect the morphology of DN3s based on staining data (Supplementary Fig. 10d), it partially and significantly blocked the heat-induced calcium reduction of DN3s (Fig. 6d, e, Supplementary Fig. 10d–f), which confirms that DN3 activity is suppressed by heat-induced DN1a activation. Connectome analysis also suggested that DN3s are modulated by light (Fig. 6a), and we have observed that the reduction in nocturnal sleep induced by heat occurs predominantly after the onset of darkness (ZT12, Supplementary Fig. 11a–c). As a result, it is hypothesized that light could potentially suppress DN3 activity, and the sleep-promoting activity of DN3s may then switch on after lights-off, with its regulation being more influenced by inhibitory DN1a input in a temperature-dependent way.

As expected, light can universally suppress DN3 activity at midday or midnight (Fig. 6f, g). Given the natural pairing of these environmental features during the day, the suppression of DN3 activity by light and high temperature is reasonable. On the other hand, the presence of lower temperature and darkness at night would free DN3 from inhibition. The observation supports that DN3 activity exhibits a trough during the day and reaches its peak at midnight[23].

The inhibitory DN1a-DN3 circuit is likely to play a major role in determining the amount of nighttime sleep in high temperature conditions, given that LNds activity is low and DN3 activity is high during the night. Thus, the Inhibition of DN3s may result in a reduction of nighttime sleep, mimicking the effects of high temperature. Our findings indicate that at 22 °C, DN3 ablation leads to a delayed offset of

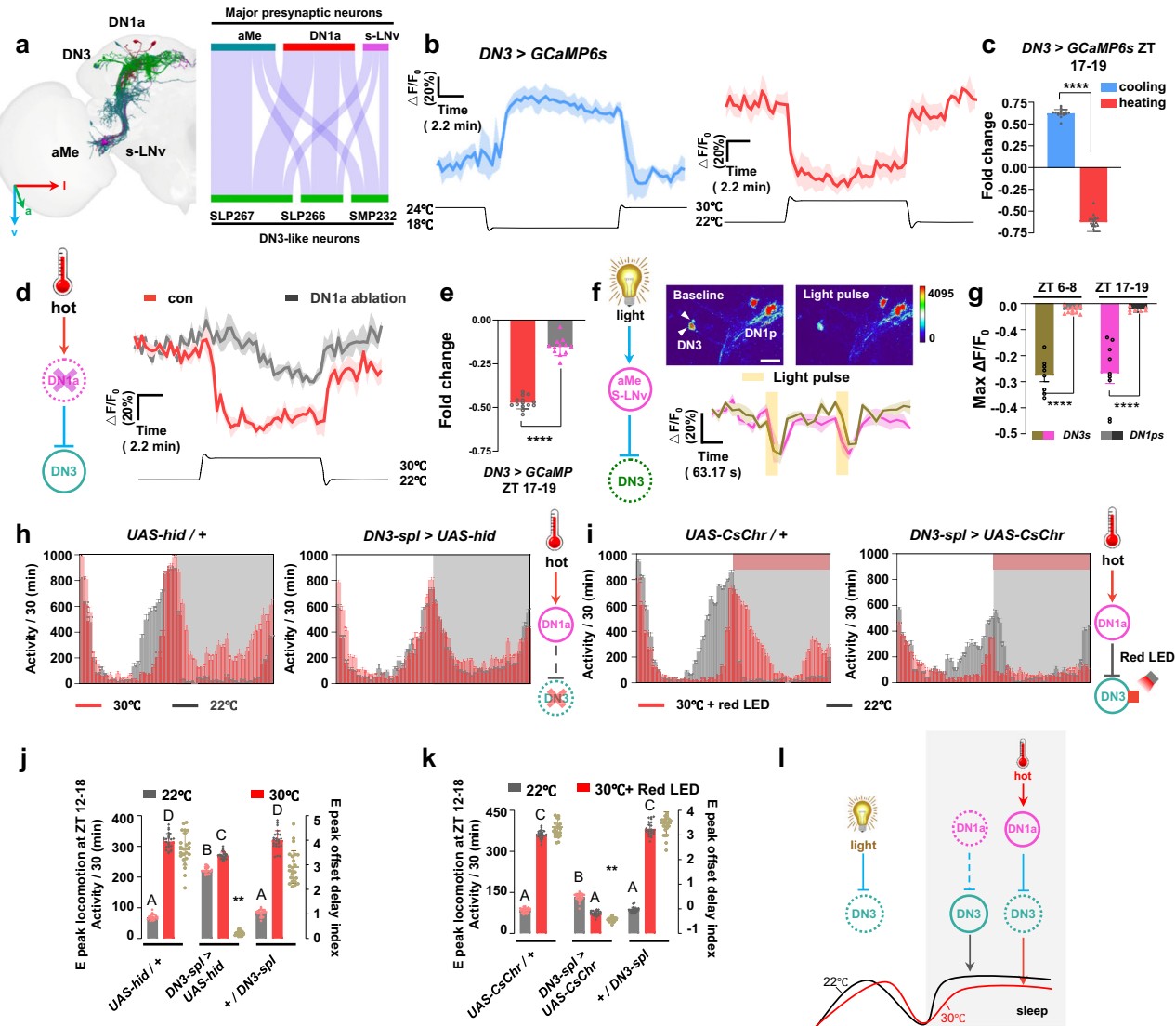

**Fig. 6 | The DN3s integrate DN1as-encoded temperature signals and light signals to modulate heat-induced E peak offset delay and nighttime sleep reduction. a** Schematic of the major upstream neurons of DN3s reconstructed in fly brain template. Left panel is the EM reconstruction of DN3s and upstream neurons. The right panel displays a Sankey histogram depicting the synaptic connections between DN3s and major presynaptic neurons. **b, d** Representative GCaMP traces ($\Delta F/F_0 \pm$ SEM) of DN3s in response to cooling (**b**, left, blue, n = 16) and heating (**b**, right, red, n = 16) in the intact fly (**d**, black, n = 12) and DN1a-silenced fly (**d**, red, n = 14). **c, e, g** Quantification of the relative fold change of calcium activities of DN3s in (**b, d, f**). Data are presented as mean in $\Delta F/F_0 \pm$ SEM in the histogram; two-tailed t-test; ****$p < 0.0001$ (**c**: $p = 9.20159\text{E-}29$; **e**: $p = 5.90626\text{E-}16$; **f**: ZT6-8: $p = 1.53564\text{E-}10$; ZT17-19: $p = 3.39847\text{E-}05$). **c** related to (**b**). **e** related to (**d**). **g** related (**f**). **f** Representative pseudocolored images of calcium responses of DN3s before and after light pulse (top). Representative GCaMP traces ($\Delta F/F_0 \pm$ SEM) of DN3s in response to light pulse at ZT 6-8 (blue, n = 8) and ZT17-19 (green, n = 9). White arrows indicate DN3s; Scale bars, 50 μm. **h, i** Representative locomotor activity profiles of control (n = 24), DN3-ablated (**h**, n = 24) and DN3 optogenetic activation (**i**, n = 24) flies at 22 °C (gray column) and 30 °C (red column). Activity summed in 30-min bins were plotted. Bar graphs are averages ± SEM; Dark shades indicate lights off. **j, k** Quantification of E-peak locomotion (column) and E-peak offset delay index (brown plot) in (**h, i**) at 30 °C (red) and at 22 °C (gray). Data are presented as mean in $\Delta F/F_0 \pm$ SEM; Statistical analysis was conducted using One-Way ANOVA followed by Tukey post-test for multiple comparisons. The letters A, B, C and D above the histograms denote significantly different means within each of the two groups, $p < 0.05$. Specific $p$ values corresponding to this figure are reported in the Source Data. **l** A model of DN3s control temperature-induced nighttime wakefulness by receiving ambient temperature information through DN1as. Dark shades indicate lights off (night).

the E peak (Fig. 6h, j, Supplementary Fig. 11a, c, d, e) and a reduction in nighttime sleep (Supplementary Fig. 11b, f), features typically observed in control flies only at 30 °C. Furthermore, the absence of further E peak offset delay or nocturnal sleep reduction in DN3-silenced flies when the temperature was increased from 22 °C to 30 °C suggested that these flies already exhibit behaviors characteristic of high temperature conditions.

The sleep characteristics in DN3 ablation flies' contrast with those in DN1a ablation flies. The latter show no delay in E peak offset and little reduction in sleep even at high temperature. Our study proposes that

DN1a inhibits DN3 at high temperature. Therefore, artificially activating DN3s at night can reverse the heat-induced increase in locomotion. To test this hypothesis, we exposed flies to hot conditions while optogenetically stimulating DN3s at night. As predicted, DN3 activation immediately induced nocturnal sleep and eliminated the heat-induced increase in locomotor activity (Fig. 6i, k, Supplementary Fig. 10g–l). Our findings suggested that high nighttime temperatures activate DN1as, leading to the inhibition of sleep-promoting DN3s and a consequent increase in locomotion and a decrease in nocturnal sleep (Fig. 6l).

## The DN1a-DN3 circuit integrates temperature fluctuation to dynamically regulate nocturnal locomotion

Our behavioral data suggest that the DN1a-DN3 circuit is involved in mediating the heat-induced increase in nocturnal locomotion. Next, we would like to directly examine the change in the activity of the DN1a-DN3 circuit during nocturnal temperature fluctuations in live flies.

To do this, we generated a *DN1a (R23E05-GAL4) + DN3-spl* that labels both DN1as and the majority of DN3s (Supplementary Fig. 12a). We performed two-photon calcium imaging with a temperature control system in a head-fixed fly entrained at ZT17-19. The fly's locomotion was continuously monitored on an air-supported treadmill. This allowed us to monitor DN1a-DN3 calcium dynamics in real-time and simultaneously record the behavioral state of the fly under temperature changes. We found that lower temperature caused a decrease in the calcium signal of DN1as, while increasing the signal in DN3s, which is consistent with our functional imaging data suggesting a role for DN1as in inhibiting DN3s (Fig. 7a, b, Supplementary Fig. 12b). Interestingly, we observed that fly locomotion was inhibited when the calcium signal from DN3s increased immediately after a rapid decrease in temperature, suggesting that low temperature signals are directly transmitted by DN1a to DN3s, resulting in reduced locomotion at low temperature (Supplementary video. 4). Conversely, when an increase in temperature resulted in the activation of DN1as, followed by inhibition of most DN3s and excitation of a small subset of DN3s (Fig. 7c, d, Supplementary Fig. 12b), we observed a significant increase in locomotion (Supplementary video. 5). These results suggest that DN1as release glutamate transmitters at high temperature, which not only inhibit GluClα⁺ DN3s but may also excite certain NMDAR1⁺ DN3s, providing a putative explanation for the previous finding of DN3 excitation by high temperature[31]. Our results also highlighted the role of DN3s as a crucial hub for the integration of DN1a-modulated temperature signals and aMe-mediated light signals to regulate E peak offset and nocturnal sleep (Fig. 7e).

## Discussion

In nature, organisms have developed a sophisticated neuronal mechanism that allows them to finely adjust their metabolism and behavior in response to fluctuations in environmental temperature, occurring either daily or seasonally[2,35]. In order to unravel the intricate encoding and regulation of temperature-dependent physiological activity changes in the central brain, we engineered a highly precise temperature-controlled Peltier device that was seamlessly integrated with an in vivo two-photon calcium imaging system. This innovative setup enabled us to capture and analyze the intricate responses of the circadian circuit to temperature variations with greater precision. Through our comprehensive investigations, we made a groundbreaking discovery: the existence of a distinct circuit encompassing DN1as, LNds, and DN3s, which acts as an integrator of both circadian and temperature information, ultimately facilitating the adaptation of fly behaviors to ambient temperature fluctuations. Notably, we unveiled the pivotal role of DN1a in regulating the activity of the locomotor-promoting LNds during the evening, as well as the sleep-promoting DN3s during the night. This exquisite regulation ensures the promotion of locomotion during cold periods and the suppression of sleep during hot nights, thereby orchestrating an optimal behavioral response to temperature changes (Fig. 7e).

Our comprehensive study extends previous findings on thermal response features in *Drosophila* central circadian neurons[7,13,16,17], as we found that DN1as respond to both increases and decreases in temperature in a circadian rhythm-dependent manner. This observation aligns with the connectome of DN1as, as their upstream neurons comprise not only TPN-IIs that convey low-temperature information but also TPN-IVs that transmit high temperature information. TPN-IV has been reported to excite LPN in response to high temperature[15],

suggesting that it should similarly activate DN1as under high temperature conditions. However, the activation of DN1as by high temperature may also be circadian rhythm-gated, and may only be detectable at certain times of day, such as nighttime. An alternative explanation is that the temperature increase method we employed differed from that of prior studies. Specifically, we developed a novel metal flyplate that was connected to a temperature-controlled Peltier device to simultaneously heat the saline in the metal dish and the air around the fruit flies. This approach better approximates how fruit flies perceive temperature in their natural environment by using the flies' antennae and peripheral temperature-sensing neurons to transmit ambinet air temperature changes to the brain.

In our observations, a response to temperature changes was evident in a subset of DN1p and DN2 neurons (Fig. 1d−g, Supplementary Fig. 1j−l). However, it is noteworthy that the response in DN1p and DN2 is less pronounced than that in DN1a during short-term temperature changes. Therefore, we did not extensively investigate this particular response. Recognizing the heterogeneity of DN1ps in terms of morphology, connectome, transcriptome, and function[13,18,29], future experiments will require more specific DN1p split-GAL4 lines to identify the temperature-sensitive subset of DN1p and explore the associated input and output circuits. Furthermore, our results do not exclude the possibility that DN1ps respond to sustained temperature changes over longer time scales. This is supported by Jin et al.'s findings, where prolonged heating at 29 °C led to a gradual increase in calcium levels in a subset of DN1ps. Moreover, potential connections between other clock neurons including DN1ps may be also involved to shape the daily calcium oscillation pattern of LNds and DN3s.

The DN1a-DN3 circuit's role in temperature-dependent sleep regulation is a crucial area of research. DN3s, an unidentified subpopulation representing approximately 35%-40% of circadian neurons, are involved in sleep regulation[30,36]. Using DN3-specific Split-GAL4 lines, we discovered that DN3s (SMP232), located downstream of DN1a, regulate temperature-dependent nocturnal sleep. Interestingly, DN3s respond to both light and high temperature, aligning with their coupling during the day. In contrast, darkness and low temperature at night release DN3s from inhibition and gradually increase their activity, consistent with previous findings showing DN3 activity peaking early in the night. DN3s are implicated in regulating the offset of the E peak and initiating nighttime sleep[23]. Additionally, DN3-secreted AstC inhibits LNds through AstC-R2, modulating the timing of evening locomotor activity[31]. Different DN3 subpopulations exhibit varying responsiveness to temperature, potentially due to the expression of specific receptors. DN3s downstream of DN1as rely on the glutamate-GluCl[31] pathway, while DN3s downstream of other temperature-sensing neurons may be activated by acetylcholine or NMDAR receptors[20]. Ablation of specific DN3 groups resulted in a delayed E peak and reduced nighttime sleep, similar to high temperature conditions. These findings demonstrate that DN3s integrate temperature, light, and circadian information to regulate physiological behaviors, such as locomotion and sleep, in fruit flies[18].

While we have identified the DN1a-LNd and DN1a-DN3 circuits that regulate physiological activities cycles in response to temperature changes in flies (Fig. 7e), we have yet to delve into the intricate molecular mechanisms through which circadian proteins exert their influence on the temperature response of circadian neurons. We propose that circadian proteins may exert their regulatory effects through multiple pathways. One plausible mechanism involves the regulation of transcription in downstream genes, such as GABA and acetylcholine receptors, to potentially enhance the sensitivity of DN1a to temperature cues mediated by GABAergic or cholinergic TPNs during nocturnal periods. Alternatively, the circadian clock may modulate diurnal variations in neuronal membrane excitability and firing patterns, as well as internal endoplasmic reticulum calcium stores, thereby influencing the diurnal calcium level and temperature response of

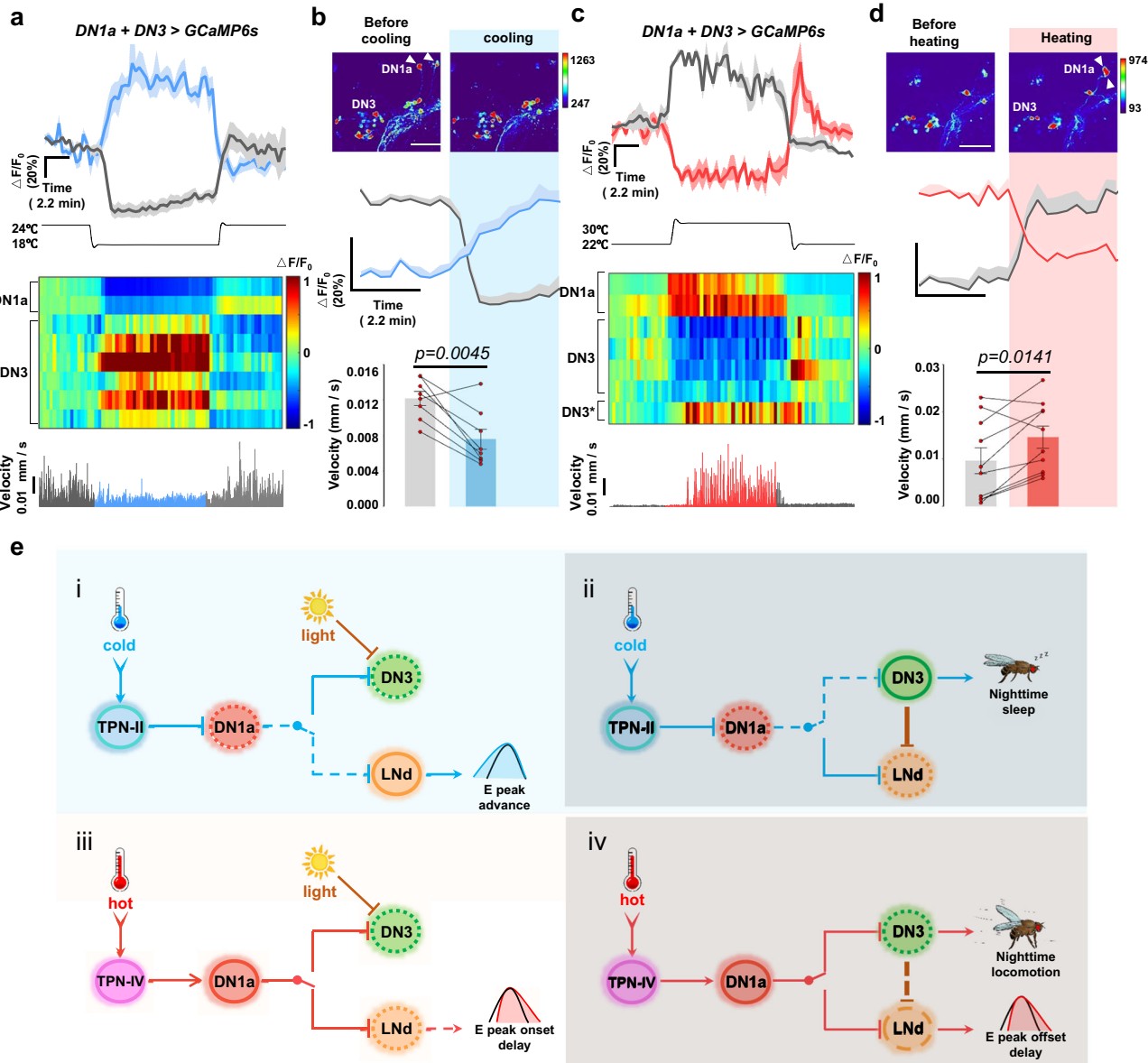

**Fig. 7 | Temperature fluctuation is integrated into the DN1a-DN3 circuit to dynamically control nocturnal locomotion. a, c** Above, the representative GCaMP traces ($\Delta F/F_0 \pm$ SEM) of DN3s (blue or red line) and DN1as (gray line) in response to cooling (**a**, $n = 8$) and heating (**c**, $n = 9$); The temperature change was labeled under the panel; Middle, heat map of the temporal calcium activities of DN3s and DN1as to cooling (24–18 °C). Below, representative recordings of movement of a tethered experimental fly walking on an air-supported ball. **b, d** Above, Representative pseudocolour images of calcium responses of *DN1a + DN3-spl>UAS-GCaMP6s* flies before and after cooling. White arrows indicate DN1as. Middle, Quantification of GCaMP traces ($\Delta F/F_0 \pm$ SEM) of DN3s (blue or red line) and DN1as (gray line) in response to short-term cooling (**b**, $n = 8$) and heating (**d**, $n = 10$). Below, the quantification of fly movement before and after cooling or heating; Data

are presented as mean $\pm$ SEM; paired samples $t$ test. Blue shadow indicates cooling; red shadow indicates heating. Scale bar, 50 μm. **e** A schematic summary of the state changes in the DN1a-LNd and DN1a-DN3 circuits that regulate temperature-induced sleep-wake changes. (**i**) Under low temperature conditions, DN1as are inhibited, leading to an early onset of the E peak. Consequently, LNd activation occurs during the daytime. In contrast, DN3s are suppressed in the presence of light. (**ii**) Inhibition of DN1as by cold night will excite DN3s to induce more nocturnal sleep. LNds will also be inhibited via the DN3-LNd circuit. (**iii**) High temperature could activate DN1as and then inhibit LNds, which may contribute to the delay in E peak onset. (**iv**). DN3s could still be inhibited by high temperature during the night via the DN1a-DN3 circuit. LNds receive less inhibitory input from suppressed DN3s to produce the delay in E peak offset. Dark shading indicates lights off (night).

DN1as[23,37]. Our data show that knockdown of SERCA, the sarco/endoplasmic reticulum calcium pump, not only reduces the amplitude of calcium rhythms in DN1a, but also enhances its diurnal responsiveness to temperature changes (Supplementary Fig. 3, 4). Nevertheless, further investigations are warranted to elucidate the precise mechanisms by which the circadian clock modulates SERCA expression levels and endoplasmic reticulum calcium levels, and to elucidate their role in regulating circadian differences in the temperature response of circadian neurons.

## Methods

### Drosophila melanogaster

All experiments were conducted using the fruit fly species *Drosophila melanogaster*. The transgenic lines employed in this study are listed in the key resources table, and the genotypes corresponding to each figure are documented in Supplementary Table 1. The flies were raised on a cornmeal-yeast-sucrose food mixture (recipe four from the Bloomington *Drosophila* Stock Center) under a 12-h light to 12-h dark cycle, with a humidity level maintained at 60–70% and a temperature

of 24 °C. The following flies were either purchased from the Bloomington *Drosophila* Stock Center or previously described and utilized in the study: *DN1a (R67F03-GAL4.AD; VT002670-GAL4.DBD), R23E05-GAL4* (ref. 21.), *Clk856-GAL4* (ref. 7.), *UAS- GCaMP6s* (ref. 18.), R67F03-LexA, UAS-tdTomato (ref. 38.), LexAop-GFP (ref. 34.), UAS-P₂X₂ (ref. 34.), *Dvpdf-LexA* (ref. 39.), *LexAop-GCaMP7s* (ref. 40.), *UAS-hid* (ref. 41.), *UAS-GtACR* (ref. 42.), DN3-spl(R67F03-GAL4.AD; VT002670-GAL4.DBD), UAS-CsChrimson.mVenus (ref. 43.), UAS-syt::GFP, UAS-DenMark (ref. 44.), UAS-Stinger-GFP, LexAop-tdTomato (ref. 45.), QUAS-GCaMP3 (ref. 46.), UAS-RFP, LexAop2-GFP (ref. 47.), QUAS-FLP (ref. 48.), Clk4.1M-LexA (ref. 49.), R18H11-LexA (ref. 47.), UAS-TNT (ref. 50.), UAS-vglut RNAi (ref. 51.), UAS-Kir2.1 (ref. 52.), UAS-TrpA1 (ref. 53.), UAS-Serca RNAi (ref. 54.), UAS-SK DN (ref. 55.), UAS-Nocte RNAi (ref. 54.) UAS-Na^{har} RNAi (ref. 26.).

## Peltier setup
The temperature of the Peltier element was measured using a thermistor, and a closed-loop feedback system employing proportional-integral-derivative control was utilized to regulate it. Both the long-term calcium monitor and live imaging experiments employed the same heating and cooling stimuli. The temperature control device (SLD70) was purchased from Sunny Precise Instruments (Shanghai) Co., Ltd. The metal flyplate[56] (Fig. 1b) was modified by Flyplate5[57]. For the live imaging tests, the fly was exposed to the surrounding air. Due to thermal resistance between the metal fly plate and the Peltier element, the actual temperature experienced by the fly slightly deviates from that of the Peltier element (Supplementary Fig. 1a). This temperature change is achieved in approximately 80 s. However, during the GCaMP trials, where the fly was submerged in saline, the thermal resistance was significantly reduced, resulting in a closer alignment between the AHL's temperature and that of the Peltier element during the heating and cooling processes (Supplementary Fig. 1a). Additionally, the time and amplitude of the heating or cooling steps during all experiments are consistent.

## Immunostaining
Adult flies were dissected in PBS, then fixed for an additional 60 min on the ice using 4% paraformaldehyde and 0.008% Triton X-100 in PBS. After three rounds of washing in PBST (PBS + 0.5% Triton-X), samples were incubated with primary antibodies at 4 °C for one or two days. After a PBST wash, secondary antibodies were applied to the brains, which were then incubated at room temperature for two hours. Samples were cleaned with PBST before mounting in VECTASHIELD Mounting Medium (Vector Laboratories) and being magnified 20 times using an Olympus laser scanning confocal microscope to see them in 1 or 1.5 µm slices. Chicken anti-GFP (Abcam, 1:1000), mouse anti-Brp (NC82) (DSHB, 1:30), rabbit anti-DsRed (Clonetech, 1:200), mouse anti-GFP (Sigma, 1:200) and rabbit anti-PER (1:1000) were the primary antibodies utilized in this study. Flies were dissected at ZT 21- and 5-days following LD entrainment for PER staining.

## Feeding of ATR
A 100 mM stock solution of all-trans-retinal (ATR) powder (Sigma) was prepared by dissolving it in 100% alcohol. For optogenetic experiments, 250 µl of the stock solution was mixed with 30 ml of 5% sucrose and 1% agar medium to prepare food with a final concentration of 400 µM ATR. Flies aged between 3 and 5 days were transferred to ATR food for a minimum of 3 days before performing optogenetic experiments[18].

## Sleep recording
The temperature-controlling apparatus (Supplementary Fig. 6a) included a metal 96-well plate (https://github.com/lihailiang7794/metal-96-well-plate), a semiconductor cooling apparatus (TEC1-12706), and a water-cooling apparatus. 96-well white Microfluor 2 plates (Fishier) with 400 µl of food (5% sucrose and 1% agar with or without 400 µM ATR) were loaded with adult male flies (aged 3–7 days). The fly movement was monitored using a camera at 10-s intervals, and the data were then used by the sleep and circadian analysis MATLAB program (SCAMP) to analyze sleep[58,59]. At least three times each of the tests were repeated. Flies were entrained to the 12 h:12 h LD cycles for three days at 24 °C or 22 °C before being exposed to 18 °C or 30 °C for one day in tests requiring temperature shift. The fourth day saw a temperature rise or drop. Flies were also entrained for three days on ATR meal to the 12 h:12 h LD cycles for optogenetic investigations. On the fourth day, channelrhodopsins like CsChrimson were activated using 5 Hz, 627 nm red-light pulses from LEDs (0.1 mW mm⁻²).

## Two-photon functional calcium imaging
Fly anesthesia was induced using $CO_2$. The head was affixed to a specialized chamber with UV adhesive, while the body was supported by a stream of air. A fine incision was made in the anterior side of the cuticle using sharp forceps, with AHL solution (108 mM NaCl, 5 mM KCl, 2 mM $MgCl_2$, 2 mM $CaCl_2$, 4 mM $NaHCO_3$, 1 mM NaH2PO₄, 5 mM HEPES, 10 mM Sucrose, 5 mM Trehalose, pH 7.5) in place. To stabilize the brain for imaging, we removed muscle tissue and cleared the tracheal passage. Only flies exhibiting vigorous movement post-operation were selected for subsequent imaging.

Prior to the experiment, flies were acclimated on the spherical treadmill for a minimum of five minutes using the two-photon Olympus FV1200MPE microscope objective. The cuticle window was exposed to AHL for ATP perfusion activation. For imaging neuronal activity during temperature changes, we captured a brain volume of 250 µm × 125 µm × 75 µm through time-lapse Z-series imaging with 25 optical sections per time frame. Fly motions on the ball were monitored and recorded using a webcam. The ball rotation speed was determined using FicTrac software[60]. Additionally, the $F_0$ setting for all flies remained constant and uniform throughout the imaging session.

To analyze the calcium traces, the Turboreg algorithm and CalmAn were used to correct for any brain motion present in the raw videos[61,62]. Fiji was used to examine ROIs. $\Delta F/F_0 = (F_t - F_0)/F_0 \times 100\%$ and $\Delta F_{Max}/F_0 = (F_{tMax} - F_0)/F_0 \times 100\%$ were used to determine the fluorescence change, where Ft is the fluorescence at time point *n* and $F_0$ is the fluorescence at time 0.

## Long-term intracellular calcium measurements
3- to 5-day-old *R14F03-GAL4.AD ∩ VT002963-GAL4.DBD > UAS-tdTomato; UAS-GCaMP7s* flies were collected under a 12:12 h LD cycle at 24 °C. The flies were then divided into four groups, each subjected to a 3-day LD entrainment with a 6-hour shift (Supplementary Fig. 2a).

On day 4, flies from the four groups, each entrained at different circadian time points, were glued into custom chambers for cuticle surgery to expose the DN1as for two-photon calcium imaging. For flies at ZT12-24, the surgery was performed under red light to avoid disruption of the dark cycle. Flies were habituated for at least 5 min prior to imaging. GCaMP and tdTomato signals in the target neurons were then recorded under two-photon microscopy within 2 min. After the rapid measurement, the chambers containing the fixed flies were returned to the original light-dark cycle for further entrainment. Each fly was measured repeatedly at 30-min intervals over a period of 6 h. Importantly, flies remained alive after the 6-hour experiments, and we ensured a minimum of 6 flies for data from each circadian time point.

We used ImageJ to calculate GCaMP and tdTomato signals using the same region of interest (ROI). The fluorescence signals from both the GCaMP channel and the tdTomato channel within the identical ROI were summed in Fiji/ImageJ. To normalize the signal, the GCaMP intensity was divided by the tdTomato intensity. The normalized calcium signals from the four groups were then plotted together to create a 24-h curve.

## E peak advanced index

An E peak advanced index was used to quantify the cooling-induced E peak advanced. It was estimated as follows: Calculate the difference between the average ZT6-12 movement on days with 18 °C and 24 °C, E peak advanced index = (average ZT6-12 movement on days with 18 °C − average ZT6-12 movement on days with 24 °C)/average ZT6-12 movement on days with 24 °C. The E peak advanced index was calculated by averaging the fly's E peak locomotion, and the experimental genotypes and matching controls were compared using one-way ANOVA and Tukey's honestly significant difference testing. E peak locomotion begins earlier when the E peak advanced index is greater.

## E peak offset delay index

The E peak offset delay index was used to quantify the heating-induced E peak offset delay. It was estimated as follows: Calculate the difference between the average ZT 12−18 movement on days with 22 °C and 30 °C, E peak advanced index = (average ZT12−18 movement on days with 30 °C − average ZT 12-18 movement on days with 22 °C)/average ZT 12−18 movement on days with 22 °C. The E peak advanced index was calculated by averaging the fly's E peak locomotion after daytime, and the experimental genotypes and matching controls were compared using one-way ANOVA and Tukey's honestly significant difference testing. E peak locomotion is finished later when the E peak offset delay index is greater.

## Nighttime sleep reduction index

The nighttime sleep reduction index was used to quantify the heating-induced sleep reduction. It was estimated as follows: Calculate the difference between the average ZT 12-24 sleep on days with 22 °C and 30 °C, nighttime sleep reduction index = (average ZT 12-24 sleep on days with 22 °C − average ZT 12−24 sleep on days with 30 °C)/average ZT12−24 sleep on days with 22 °C. The nighttime sleep reduction index was calculated by averaging the fly's sleep during the night. Experimental genotypes and matching controls were compared using one-way ANOVA and Tukey's honestly significant difference test. Heat-induced nighttime locomotion is increased when the nighttime sleep reduction index is greater.

## E peak sleep reduction index

The E peak sleep reduction index served to quantify heating/cooling-induced sleep reduction during the evening. In cold temperature conditions, it was determined as follows: calculate the difference between the average ZT 6−12 sleep on days with temperatures at 18 °C and 24 °C. The E peak sleep reduction index is then expressed as (average ZT 6-12 sleep on days with 18 °C − average ZT 6−12 sleep on days with 24 °C)/average ZT6-12 sleep on days with 24 °C.

For hot temperature conditions, the index was estimated by calculating the difference between the average ZT 12-18 sleep on days with temperatures at 18 °C and 24 °C. The nighttime sleep reduction index is expressed as (average ZT 12−18 sleep on days with 30 °C − average ZT 12-18 sleep on days with 22 °C)/average ZT12-18 sleep on days with 22 °C.

The E peak reduction index was calculated by averaging the fly's sleep in the evening. Experimental genotypes and their respective controls were compared using one-way ANOVA and Tukey's honestly significant difference test. An increase in heat-induced nighttime locomotion corresponds to a higher E peak sleep reduction index.

## Light stimulation experiment

Male flies aged 3−5 days were maintained for three days under a 12:12 LD (light-dark) cycle at 24 °C. On the fourth day, they were glued during a specific time window and subjected to two-photon calcium imaging. During the imaging process, a 1 Hz, 0.1 mW/mm² white LED light stimulus was applied. It's important to note that when dissecting flies at ZT17-18, this procedure was conducted under red light.

Additionally, it's crucial that the light stimulus doesn't directly illuminate the neuron cell bodies during the light stimulation phase of the experiment.

## Hemibrain connectome analysis and visualization

Hemibrain dataset v1.2.1 was used to gather information on the synaptic connections between the presynaptic DN1as and the post-synaptic neurons of interest. The connectivity was visualized with a Sankey diagram by the Plotly R Studio library (https://plotly.com/r/sankey-diagram/).

## Correlation and clustering analysis

The heterogeneity and hierarchical response within the circadian circuit were examined through correlation and clustering analysis. This analysis utilized various R Studio libraries, such as Tidyverse, Reshape2, ComplexHeatmap, and Circlize. To visualize the results, a heatmap was generated, facilitating the identification and clustering of patterns in the circadian circuit's response to temperature changes. The insights gained from this analysis shed light on the intricate dynamics of the circadian circuit and its sensitivity to fluctuations in temperature.

## Data analysis, statistics, and reproducibility

Statistical analyses were performed using IBM SPSS and Prism software. The normal distribution of the data was assessed using the Wilks-Shapiro test. For normally distributed data, two-tailed, unpaired Student's $t$ tests were conducted, along with one-way ANOVA followed by the Tukey-Kramer HSD Test as the post hoc analysis. The number of animals (n value) for each experiment was indicated in the figure legends. Behavioral responses were presented as mean values, and error bars represented the standard error of the mean (SEM). Statistical significance was determined if the p-value was less than 0.05 ($p < 0.05$). For immunostainings, the experiment was independently repeated three times, and one representative image was displayed. No data were excluded from the analysis. Randomization or blinding was not implemented during experiments or data analysis. Additionally, no statistical method was used to predetermine sample size.

## Reporting summary

Further information on research design is available in the Nature Portfolio Reporting Summary linked to this article.

# Data availability

All data, including relevant raw data from each figure (in both the main paper and Supplementary Information), are accessible within the main text or supplementary materials. A Source Data File accompanies this paper, containing the necessary raw data. The study utilized the following databases: Flylight (https://www.janelia.org/project-team/flylight), Bloomington Drosophila Stock Center (https://bdsc.indiana.edu/), Vienna Drosophila Resource Center (https://shop.vbc.ac.at/vdrc_store/), Codex (https://codex.flywire.ai/) and neuPrint (https://neuprint.janelia.org/). Source data are included with this paper. Source data are provided with this paper.

# Code availability

The scripts employed in this study have been deposited on GitHub [https://github.com/lihailiang7794][56].

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

## Acknowledgements

We thank the members of the Fang Guo Lab for their generous help, the members of the Zhefeng Gong Lab for laboratory support. We thank Dingbang Ma from Michael Rosbash Lab, Hongtong Lin, and Youshi Guo for their scientific discussion and comments on this manuscript. We also thank Dr. Zhefeng Gong, Dr. Chang Liu, Dr. Yi Rao, Dr. Yufeng Pan, Dr. Zhihua Liu, the Vienna *Drosophila* Resource Center, and the Bloomington *Drosophila* Stock Center for stocks and reagents. We thank Sanhua Fang and Dan Yang from the Core Facilities, Zhejiang University School of Medicine for their technical support. We thank Xinling Wang for providing the hand-drawn cartoons featured in this manuscript. This work is supported by funding from the National Key Research and Development Program of China (2019YFA0802400), the National Natural Science Foundation of China (32171008, 31970941), the Zhejiang Provincial Outstanding Youth Science Foundation (LR20C090001), the Non-profit Central Research Institute Fund of Chinese Academy of Medical Sciences (2023-PT310-01), the Fundamental Research Funds for the Central Universities (2023ZFJH01-01, 2024ZFJH01-01) and the Fundamental Research Funds for the Central Universities to F.G.

## Author contributions

H.L.L. and F.G. conceived and designed the experiments. H.L.L., Z.Y.L., Y.T., X.Y., WJ.Y. and P.Y.Z. performed the behavioral experiments, immunostaining, and functional imaging experiments. H.L.L. and Z.Y.L. performed the *trans*-Tango experiments. H.L.L. and Z.Y.L. analyzed the data. H.L.L. and Z.Y.L. prepared the Figures. H.L.L., Z.Y.L., X. M. L. and F.G. wrote and revised the paper.

## Competing interests
The authors declare no competing interests.
