## [Peer Review File · Nature Communications]

Dynamic encoding of temperature in the central circadian circuit coordinates physiological activitiesEditorial Note: Parts of this Peer Review File have been redacted as indicated to remove third-party material where no permission to publish could be obtained.

REVIEWER COMMENTS

Reviewer #1 (Remarks to the Author):

In this manuscript, the authors seek to understand how *Drosophila* clock neural network integrates temperature information to regulate behavior. They identified DN1a's, a group of dorsal clock neurons, as the key temperature sensitive neurons: DN1a's are inhibited by cold and activated by hot stimuli. They further characterized that the neural circuit connecting DN1a's, LNd's, and DN3 is critical for processing temperature information. Using imaging, functional, and behavioral assays, they discovered that DN1a's inhibit both activity-promoting LNd's and sleep-promoting DN3's. Specifically, they found that DN1a-LNd connection is critical for regulating the evening peak in response to cold temperature, while DN1a-DN3 connection is critical for night-time sleep reduction in response to warm temperature. Numerous prior studies, some from the same group, have delved into the temperature-sensitivity of clock neurons. While the present research builds on these previous findings, it seems to fall short in providing a deeper mechanistic understanding.

Major concerns:

1) The authors performed long-term calcium measurements and found that calcium levels in DN1a's rise during the day and decline at night. These measurements appear to have been taken under light-dark conditions at 25°C. Considering the light-dark cycle, how can the researchers assert that this fluctuation is due to temperature? If there's continuous illumination for 12 hours during the analysis, could this cause the GCaMP fluorescence signal to bleach? Paul Taghert's lab conducted long-term calcium imaging experiments under dark-dark conditions with a light-sheet microscope. How do the current study's findings align with those previously published?

2) The authors demonstrate a correlation between DN1a temperature response and PER levels. Although this is a noteworthy finding, it remains a correlational observation. Previous research has indicated that temperature or light responses in different clock neurons are influenced by the circadian clock. What underlying mechanism is responsible for this?

3) The authors showed that DN1a's suppress both the LNd's that promote activity and the DN3's that promote sleep. Their findings suggest that the intensity of these neural connections could vary throughout the day. For instance, the inhibitory link between DN1a and LNd might be more pronounced in the early evening, whereas the DN1a-DN3 inhibitory connection might dominate during the night. The mechanisms of how this is achieved is not clear. Although the influence of light on the DN1a-DN3 connection is discussed, the potential impact of light on the DN1a-LNd inhibitory relationship remains unaddressed. My major criticism is that the authors did not delve into the mechanisms of how the intensity of these neural connections could vary over the circadian cycle.

4) Considering that DN1a's suppress LNd's, can the authors clarify why the calcium levels in both DN1a and LNd's are low during the night?

Reviewer #2 (Remarks to the Author):

General comments

In this study, the authors discovered that DN1a's are most temperature sensitive neurons in central circadian circuit and show diurnal temperature responses. DN1a's target two circadian subjects, LNd's and DN3's to modulate activity and sleep in different ambient temperatures. Many techniques are employed such as 3D printing, two photon confocal imaging, EM connectome analysis and trans-Tango etc. The designs of research are straightforward. The results are convincing. However, there are some concerns need to be addressed.

Major comments:

1.

The researchers posit that DN1as exhibit the highest degree of sensitivity among circadian neurons in response to shifts in ambient temperature. This finding contradicts a previous study conducted by Yadlapalli et al. in 2018[1]. The authors attribute this discrepancy to variations in methodology. However, I note that the authors and Yadlapalli et al. conducted experiments within different time frames. Yadlapalli et al. introduced a temperature elevation to 29 °C in a span of 7 seconds, maintaining it for only 5 seconds upon light activation (as per their methods). Conversely, the authors of this study omitted precise details regarding the duration of their temperature shifting and captured the GCaMP signal at a temporal resolution of minutes.

It is essential to recognize that neurons might exhibit diverse responses to brief temperature fluctuations and to sustained temperature alterations, potentially contributing to the observed disparities, as suggested by Jin et al. in 2021[2].

Therefore, it is imperative for the authors to delineate, during their results discussion, that the conclusions drawn are applicable solely to the specific experimental conditions and temporal scales employed.

Based on the findings depicted in Fig 1d and Fig 1g, I concur with the evident changes in calcium signaling within DN1a. Nevertheless, the authors have omitted an analysis of the responses exhibited by DN1p and DN2 neurons. The outcomes indicate that distinct DN1p and DN2 neuronal populations might react differentially to temperature stimuli. This discrepancy should not be misconstrued to imply insignificance or insensitivity to thermal cues. Rather, it suggests that the underlying mechanisms are more intricate than the discourse presented by the authors in lines 379-380.

Overall, the manner in which the authors present and discuss their findings warrants a greater degree of prudence and moderation.

2.

Indicating the experimental procedures with ZT time notation would be highly beneficial, as the outcomes of the experiments could potentially be influenced by circadian rhythmic factors. For instance, specifying the ZT time at which the experiments illustrated in Figures 1d to 1h were conducted would provide essential contextual information.

3.

It is interesting that, whether the LN_d or DN3, by themselves, can respond to temperature changes, if the synaptic nerve transmission was blocked by tetrodotoxin (TTX) to rule out the effect from the circuit/ potential upstream neurons.

4.

Regarding Figure 2a, can it be confirmed if these neurons originate from the same brain? If not, how have potential inter-individual variations been accounted for? Concerning Figure 2b, could you clarify if these recordings span 24 hours continuously or occur at six-hour intervals with fly changes? Additionally, it would be helpful to know the number of flies averaged for these recordings.

An intriguing observation arises when comparing these results with prior research, such as[3]. Liang et al. employed a similar methodology and reported heightened DN1 neuron activity during the nighttime, contrasting with the present study's assertion of increased DN1a activity in the daytime. A comprehensive explanation and discussion of these discrepancies are warranted. From the technical level, for example data in Figure 2, it looks like that, lower F0 baseline values of calcium imaging are correlated with higher $\Delta F/F_0$. I am wondering if the authors fix all test group's F0 to a certain value (like 800 in 12bit depth grey recording) by adjusting confocal gain value settings, the differences among Max $\Delta F/F_0$ values would be the same conclusion.

5.

In reference to Figures 2e-g, where the author asserts a phase-locked daytime-like response in Per mutant flies (as indicated in lines 143-145), I suggest that a rescue experiment is conducted within the time window of ZT 0-2. This specific time frame aligns with the circadian system's maximal suppression for gating DN1as' responsiveness to temperature fluctuations.

6.

In Figures 6, the utilization of UAS-hid and optical genetics by the authors to validate DN1 functionality is notable. However, these two approaches possess limitations. The application of UAS-hid results in the ablation of specific components within the circadian network, potentially triggering unintended consequences such as altering the broader circadian pacemakers and yielding comprehensive outcomes. In the context of utilizing optical genetics for locomotion or sleep testing, it's important to acknowledge that light exposure can introduce confounding effects on behavior, as demonstrated by [4] and other reports.

In light of these considerations, it is recommended to explore alternative genetic tools like Kir2.1, TrpA1 or others to manipulate the neuronal activity, thereby offering a further layer of validation for the observed behavioral outcomes. The author used the TNT to block the DN1as signal, but I'd like to see more data related to genetically manipulate the downstream neurons like LNds or DN3s.

7.

In lines 374-382, given the limited presence of only two DN1a neurons on one side, it becomes evident that DN1a constitutes a minor subset of neurons characterized by the 18H11-GAL4 label. Consequently, ascertaining the precise implications generated by the use of the 18H11-GAL4 driver remains challenging. Referring to Figure 3c, it indicates that DN1p neurons are labeled by the $DN1p-GAL4 > tans-tango$. Hypothetically, utilizing the $DN1p-GAL4 > tans-tango$ driver might also result in the labeling of DN1a neurons (although this assumption warrants further investigation). It is important to note, however, that the inherent interplay within the DN1 neuronal population demands attention. Functional distinctions between DN1a and DN1p might be more complicated. In light of these considerations, I recommend that the authors exercise caution and refrain from prematurely drawing conclusive insights in line 381-382.

Minor concern:

1.

In Figure 1j, the actual ΔT are varied in 4 to 5 different switching programs, however, the authors only show the temperature switching from 24 to 18 or 22 to 30 Celsius degrees. Better to modify the temperature switching curves.

2.

There appear to be several errors or oversights in the citations. In line 45-46, the correct reference should be [5] rather than [2]. Similarly, in line 166-167, the appropriate citation is [6]. Additionally, in line 147-148, it is recommended to include a citation to [7] at minimum. As you are submitting to a comprehensive journal, please ensure to provide appropriate citations for aspects that might be considered common knowledge to you but could be challenging for other readers to comprehend.

3.

The authors should provide clarification regarding the rate at which the system achieves specific temperature increases or decreases. Additionally, it is essential to address whether the heating or cooling time remains consistent when employing different "steps," as mentioned in lines 96-97.

[1] S. Yadlapalli, C. Jiang, A. Bahle, P. Reddy, E. Meyhofer, and O. T. Shafer, "Circadian clock neurons constantly monitor environmental temperature to set sleep timing," *Nature*, vol. 555, no. 7694, pp. 98–102, Mar. 2018, doi: 10.1038/nature25740.

[2] X. Jin, Y. Tian, Z. C. Zhang, P. Gu, C. Liu, and J. Han, "A subset of DN1p neurons integrates thermosensory inputs to promote wakefulness via CNMa signaling," *Current Biology*, vol. 31, no. 10, 2021, doi: 10.1016/j.cub.2021.02.048.

[3] X. Liang, T. E. Holy, and P. H. Taghert, "Synchronous drosophila circadian pacemakers display nonsynchronous Ca^{2+} rhythms in vivo," *Science* (1979), vol. 351, no. 6276, 2016, doi: 10.1126/science.aad3997.

[4] D. D. Au et al., "Drosophila photoreceptor systems converge in arousal neurons and confer light responsive robustness," *Front Neurosci*, vol. 17, 2023, doi: 10.3389/fnins.2023.1160353.

[5] A. Lamaze, A. Öztürk-Çolak, R. Fischer, N. Peschel, K. Koh, and J. E. C. Jepson, "Regulation of

sleep plasticity by a thermo-sensitive circuit in *Drosophila*," *Sci Rep*, vol. 7, 2017, doi: 10.1038/srep40304.

[6] S. M. Plaza et al., "neuPrint: An open access tool for EM connectomics," *Front Neuroinform*, vol. 16, 2022, doi: 10.3389/fninf.2022.896292.

[7] J. L. Price, M. E. Dembinska, M. W. Young, and M. Rosbash, "Suppression of PERIOD protein abundance and circadian cycling by the *Drosophila* clock mutation *timeless*," *EMBO Journal*, vol. 14, no. 16, 1995, doi: 10.1002/j.1460-2075.1995.tb00075.x.

Reviewer #3 (Remarks to the Author):

The Guo lab shows an interesting topic about temperature representation in the clock circuits. They employ a fascinating new technique with a two-photon microscope, which allows them to map the circadian clock circuits by analyzing the neural activity to temperature. Although several researches suggest that temperature is represented in the clock circuits, the details are still largely vague. The Guo group found that DN1as are activated/inhibited by warm/cold temperatures. DN1as targets two circuits, E cells (activity) and DN3s (sleep). Overall, it is a very interesting paper, but I have several comments that can help improve the quality of this paper.

Main comments:

1. Discrepancies with the Nature paper (Yadlapalli et al., 2018)

There are several discrepancies from the Nature 2018 paper. The authors briefly explain the reasons. However, they need to show/explain more details of the differences between this study and the Nature 2018 paper and discuss why it is different. For example, please describe each cell's activity against warm/cold temperatures and how this data differs from Nature 2018. It would be great if a summary of the differences is provided. The author mentioned a possible difference related to the reporters they used. I wonder if there are any scientific reasons behind this. It sounds like a non-evidence based.

Lane 95: "Overall, our study provides evidence DN1as can detect rapid changes in ambient temperature,"

Why does the author think DN1as can detect "rapid" changes compared to Nature 2018? Is this based on the experiments? If so, please describe it in more detail.

Also, please discuss the reasons for the "rapid" temperature response. What is the benefit of the fly's physiology?

2. Sleep data

The authors discuss sleep phenotype using locomotor activity data, but these are not perfectly equal. They should use sleep data instead of locomotor activity data for evaluating the sleep phenotypes.

3. Overstated explanations

I feel there are several overstated explanations. The author's interpretation is beyond what the actual data meant. I did not perfectly agree with some of the author's description/explanation. Here are some examples:

Line 113: "Our findings revealed a rise in calcium levels during the day and a decline at night, which corresponds to natural daily temperature changes (Fig. 2a, b). These observations suggest that DN1a may have evolved in the fly brain as a circadian thermometer to accurately reflect daily environmental temperature."

The condition of this experiment was LD cycles at the constant temp, and DN1a shows the fluctuation of Ca levels. It seems not to be the response to the ambient temp but light-dark cycles. I do not agree the author's statement. Please confirm each explanation/statement is non-biased and non-overstated.

Lane 146: "This data supported our hypothesis that the circadian clock gated the temperature response, as per mutant flies mimic the daytime conditions in which wild-type flies have low PER levels in circadian neurons."

I understand that the circadian clock gated the temperature response. When PER is expressed more, the neural activity is lower, but why do the authors mention it is related to the PER level? Is there any scientific evidence? It may be just a coincidence.

4. Missing information about the experiments

Line 541: Long-term Intracellular Calcium Measurements

How does in vivo long-term intracellular calcium measurements are measured? Also, it is not clear how to measure it under the light. I can't find the details in the material and method or other places.

Fig. 2 legend, Lane 623: "the results are shown in the histograms labeled A, B, C, D, E and F." The statistical analysis is unclear.

Lane 645 g, h (g): "Quantification of the relative fold change of calcium activities of DN1as in (e-f). (h): Basal GCaMP6s signals in DN1as at two circadian time points in (e-f). Data are presented as mean in $\Delta F/F_0$ (%) \pm SEM in the histogram; Tukey's honest significant difference tests were used to determine the statistical significance of the means within each experimental group, and the results are shown in the histograms labeled A, B, C and D. Pairwise comparisons were made, and all differences between means were significant at $P < 0.05$."

The statistical analysis is unclear. What "baseline calcium level" in (h) is?

Lane 291 "As expected, light can universally suppress DN3 activity at midday or midnight (Fig. 6f, g)."

The experimental procedure is unclear. How did the authors test the effect of the light in ZT6-8? Fig. 6 figure legend is unclear. h. The explanation is poor and hard to understand, and the figure legend for (g) is missing.

Minor comments:

Lane 33: 11 and 12 references do not include the feeding behavior.

Lane 175: "The DN1a-E cell circuit is an intriguing focus of study because the LNds and 5th s-LNV are the primary neurons innervated by DN1as, and these E-cells control the evening locomotion peak, which is dynamically reshaped by temperature."

There is no citation.

Lane 213: "As noted earlier, the calcium levels in LNds exhibit a peak during the evening, and the regulation of the onset and offset of the evening locomotor peak, which is under the control of LNds, can be influenced by temperature variations."

Which data is talking about?

Lane 273; DN3-like neurons (SLP266, SLP267, and SMP232)

Please explain more. What are they?

Lane 354: "Notably, we unveiled the pivotal role of DN1a in regulating the activity of the locomotor- and feeding- promoting LNds during the evening, as well as the sleep-promoting DN3s during the night. This exquisite regulation ensures the promotion of locomotion and feeding during cold periods."

Where the "feeding information" is coming from?

Lane 274: "Additionally, the data indicated that DN1as and light input neurons, such as aMe and s-LNv, potentially exert a significant impact on DN3-like neurons, which implies a probable correlation between DN3 activity and its modulation by light and temperature."

The reference is missing.

Lane 285: "Connectome analysis also suggested that DN3s are modulated by light, and we have observed that the reduction in nocturnal sleep induced by heat occurs predominantly after the onset of darkness (ZT12)."

Which data is talking about?

We extend our sincere gratitude to the three reviewers for their invaluable comments and constructive suggestions. In our revised submission, we have taken meticulous care to address each point raised by the reviewers, with the specific alterations to the manuscript thoughtfully highlighted in color. These revisions have undoubtedly enhanced the clarity and quality of the paper. We are delighted to observe that all three reviewers share our enthusiasm for the research findings presented in our manuscript. We firmly believe that our work offers novel perspectives and makes a valuable contribution to the existing literature.

Reviewer #1 (Remarks to the Author):

In this manuscript, the authors seek to understand how Drosophila clock neural network integrates temperature information to regulate behavior. They identified DN1a's, a group of dorsal clock neurons, as the key temperature sensitive neurons: DN1a's are inhibited by cold and activated by hot stimuli. They further characterized that the neural circuit connecting DN1a's, LNd's, and DN3 is critical for processing temperature information. Using imaging, functional, and behavioral assays, they discovered that DN1a's inhibit both activity-promoting LNd's and sleep-promoting DN3's. Specifically, they found that DN1a-LNd connection is critical for regulating the evening peak in response to cold temperature, while DN1a-DN3 connection is critical for night-time sleep reduction in response to warm temperature. Numerous prior studies, some from the same group, have delved into the temperature-sensitivity of clock neurons. While the present research builds on these previous findings, it seems to fall short in providing a deeper mechanistic understanding.

We are grateful for the invaluable comments offered by the reviewer. As outlined below, we have diligently revised the manuscript to enhance its precision and informativeness. We firmly believe that these revisions hold significant importance in refining the quality of the manuscript.

Major concerns:

1) The authors performed long-term calcium measurements and found that calcium levels in DN1a's rise during the day and decline at night. These measurements appear to have been taken under light-dark conditions at 25°C. Considering the light-dark cycle, how can the researchers assert that this fluctuation is due to temperature?

Thank you for your valuable comment. First, we apologize for the lack of clarity in our description. The long-term in vivo calcium imaging was performed at a constant temperature of 24 degrees under LD conditions, so the calcium fluctuations are not caused by temperature changes. Our original description may have been misleading and we have now revised this section.

We think that the fluctuations in calcium signaling in DN1a are regulated by the circadian clock. The data in Figures 2a-b of the manuscript show that calcium levels in DN1a are higher during the day and lower at night, which is similar to the accumulation and degradation pattern of circadian proteins. In addition, the data in Figures 2e-h of the manuscript show that in *per0* flies, DN1a has higher basal calcium levels both day and night, whereas in DN1a>PER rescue flies, PER levels were overexpressed, and in this scenario, DN1a basal calcium levels were always lower both day and night. Taken together, manipulation of PER levels in DN1a can dramatically alter the basal circadian activity of DN1a. Therefore, calcium oscillations in DN1a are more likely to be regulated by the endogenous circadian clock.

We have revised the original text to state: “Our findings unveil a diurnal pattern in the calcium levels of DN1as, characterized by an increase during the day followed by a decrease at night . This rhythmic fluctuation of DN1a calcium levels aligns with the natural light-dark cycle and suggests potential regulation by an intrinsic circadian clock.”

If there's continuous illumination for 12 hours during the analysis, could this cause the GCaMP fluorescence signal to bleach?

Thanks to the helpful comments, it appears that our previous description lacked

clarity. To provide further clarification, during our analysis, we did not conduct continuous imaging of flies for 12 hours. Instead, we measured DN1a calcium levels in the same flies glued onto imaging chambers every 30 minutes, with each imaging session lasting less than 2 minutes. The flies with chambers were then returned to their original LD cycle in the incubator. The entire procedure was completed within a 6-hour timeframe. We have now clearly outlined our long-term imaging protocol in the Methods section:

“3- to 5-day-old *R14F03-GAL4.AD* \cap *VT002963-GAL4.DBD* > *UAS-tdTomato*; *UAS-GCaMP7s* flies were collected under a 12:12 hours LD cycle at 24 °C. The flies were then divided into four groups, each subjected to a 3-day LD entrainment with a 6-hour shift, as detailed below.

On day 4, flies from the four groups, each entrained at different circadian time points, were glued into custom chambers for cuticle surgery to expose the DN1as for two-photon calcium imaging. For flies at ZT12-24, the surgery was performed under red light to avoid disruption of the dark cycle. Flies were habituated for at least 5 min prior to imaging. GCaMP and tdTomato signals in the target neurons were then recorded under two-photon microscopy within 2 minutes. After the rapid measurement, the chambers containing the fixed flies were returned to the original light-dark cycle for further entrainment. Each fly was measured repeatedly at 30-minute intervals over a period of 6 hours. Importantly, flies remained alive after the 6-hour experiments, and we ensured a minimum of 6 flies for data from each circadian time point.

We used ImageJ to calculate GCaMP and tdTomato signals using the same region of interest (ROI). The fluorescence signals from both the GCaMP channel and the tdTomato channel within the identical ROI were summed in Fiji/ImageJ. To normalize the signal, the GCaMP intensity was divided by the tdTomato intensity. The normalized calcium

signals from the four groups were then plotted together to create a 24-hour curve.”

Paul Taghert's lab conducted long-term calcium imaging experiments under dark-dark conditions with a light-sheet microscope. How do the current study's findings align with those previously published?

We appreciate the insightful comments of the reviewers. It's important to highlight the differences between our study and Paul Taghert's data, which were conducted under constant darkness conditions. Paul Taghert's investigations were aimed at studying the intrinsic rhythms of circadian neuronal activities, whereas our experiments were intentionally designed to be performed under light-dark conditions, in line with our LD behavioral data. In addition, we chose not to record the calcium activity of neurons in the same flies continuously for 24 hours under two-photon microscopy to prevent fluorescence bleaching and to preserve the health of the flies.

It's noteworthy that in Paul Taghert's paper, they didn't differentiate between DN1a and DN1p, referring to them collectively as DN1s. Given the predominance of DN1ps (~15) compared to DN1as (2) per each hemisphere, it is likely that the DN1 signal they measured primarily originates from DN1ps. To address this, we employed the DN1p-specific driver *Clk4.Im-GAL4* to assess the daily calcium fluctuation of DN1p in LD. Importantly, our DN1p daily calcium fluctuation pattern recorded by in-vivo two-photon calcium imaging method is similar to those reported by Paul Taghert, with a peak around late night. This data suggested distinct phases of calcium rhythms between DN1a and DN1p (Figure 1 of the reply).

Figure 1 Calcium oscillation patterns of DN1a and DN1p through long-term *in vivo* calcium recording in live flies.

Left: Fluorescence signal diagrams of GCaMP and tdTomato in DN1p across 4 circadian time points. **Right:** The averaged calcium fluctuation patterns of DN1a and DN1p at different time points during the day. The data was double-plotted to better visualize the pattern.

2) The authors demonstrate a correlation between DN1a temperature response and PER levels. Although this is a noteworthy finding, it remains a correlational observation. Previous research has indicated that temperature or light responses in different clock neurons are influenced by the circadian clock. What underlying mechanism is responsible for this?

Thanks to the thoughtful comments, we have revisited our previous findings, with particular attention to the data on the *per⁰* mutant and the rescue of PER (Figure 2 of the manuscript), as shown in Figures a-b below. We think that our data suggest that DN1a activity and PER levels are more than a correlation or coincidence. As in *per⁰* flies, DN1a have higher basal calcium levels both day and night, whereas in DN1a>PER rescue flies, PER levels were overexpressed (Supplementary Figure 2f-h of the manuscript), and in this scenario, DN1a basal calcium levels were always lower both day and night. Taken together, manipulation of PER levels in DN1a can dramatically alter the basal circadian activity of DN1a as well as the diurnal calcium response to environmental temperature fluctuations.

To gain a deeper understanding of the mechanism of how PER affects DN1a activity, we performed a comprehensive screen targeting potential proteins that have been reported to be associated with the circadian clock to regulate circadian neuronal activity

or temperature process, including narrow abdomen (Bridget C. Lear, et al. *Neuron*. 2005), shaker (Philip Smith, et al. *J Physiol*. 2019), nocte (Chenghao Chen, et al. *Curr Biol*. 2018), and SERCA (Liang et al. *PNAS*. 2022), as shown in Figure 13 a-d of reply. Notably, when SERCA was specifically knocked down in DN1a, we observed that DN1a lost the circadian variation of calcium levels, with significantly lower basal calcium levels in both day and night. Consequently, these flies showed a bigger response to temperature changes during day. This result is consistent with Liang's data showing that SERCA RNAi in circadian neurons can decrease the amplitudes of calcium fluctuations in circadian neurons.

In addition, our behavioral data showed that specific knockdown of SERCA in DN1a, which reduces DN1a activity, can effectively mimic a cold phenotype, specifically an E peak advance at normal temperature. These flies can further advance the E peak when temperatures are lowered.

Taken together, these data suggest a regulatory role for PER in modulating calcium rhythm in DN1a, as well as the rhythmicity of DN1a in response to temperature fluctuations, possibly through the ER calcium channel SERCA. Considering that we already have 7 main figures and more than 10 supplemental figures, the detailed mechanism will be investigated in the future. This part of the data has now been added to the Supplementary Fig. 3 in the manuscript.

Figure 2 Circadian clock modulate the DN1a diurnal calcium response to ambient temperature variations via SERCA.

a, b The representative GCaMP traces ($\Delta F/F_0$) of DN1as in response to cooling in flies at ZT 6-8 (**a**) and ZT 17-19 (**b**). green curve: wild type; red curve: *per* mutant; purple curve: PER rescue. Dark shades indicate lights off (night); $n \geq 6$.

c, d (**c**): Quantification of the relative fold change of calcium activities of DN1as in (**a-b**). (**d**): Basal GCaMP6s signals in DN1as at two circadian time points in (**a-b**). Data are presented as mean in $\Delta F/F_0$ (%) \pm SEM in the histogram; Tukey's honest significance difference test to assess

the statistical significance of the mean values within each experimental group. In the provided histograms labeled A, B, C and D, the use of the same letter denotes the absence of a significant difference between the two groups, whereas differing letters signify a $p < 0.05$, indicating a significant distinction between the groups. The baseline calcium level represents the spontaneous calcium level of DN1a before temperature fluctuations.

e, Representative locomotor activity profiles of control (n=24) and SERCA knockdown(n=24) flies at 24°C (grey column) and 18°C (blue column). Activity summed in 30-min bins were plotted. Bar graphs are averages \pm SEM; Dark shades indicate lights off (night).

f, Quantification of E peak locomotion and E peak advance index (magenta) in (e). The grey and blue column indicates flies' locomotion at 24°C and 18°C; the purple plots indicate E peak advance; Data are presented as an average locomotion \pm SEM in the histogram; Tukey's honest significance difference test to assess the statistical significance of the mean values within each experimental group. In the provided histograms labeled A, B, C and D, the use of the same letter denotes the absence of a significant difference between the two groups, whereas differing letters signify a $p < 0.05$, indicating a significant distinction between the groups. ** $P < 0.01$.

3)The authors showed that DN1a's suppress both the LNds that promote activity and the DN3s that promote sleep. Their findings suggest that the intensity of these neural connections could vary throughout the day. For instance, the inhibitory link between DN1a and LNd might be more pronounced in the early evening, whereas the DN1a-DN3 inhibitory connection might dominate during the night. The mechanisms of how this is achieved is not clear. Although the influence of light on the DN1a-DN3 connection is discussed, the potential impact of light on the DN1a-LNd inhibitory relationship remains unaddressed.

Thanks to the thoughtful comments, we revisited the LNd response to light stimulation. Illustrated in the figure below, the calcium levels of LNd exhibited no significant variations in response to light stimulation across different circadian time points during the day.

Figure 3 LNds' response to photo-stimulation.

a, Representative pseudocolored images of calcium responses of LNds before and after light pulse (top). Representative GCaMP traces ($\Delta F/F_0$) of LNds in response to light pulse at different circadian time points.

b, Quantification of the relative fold change of calcium activities of LNds in (a).

My major criticism is that the authors did not delve into the mechanisms of how the intensity of these neural connections could vary over the circadian cycle.

Thanks to the thoughtful comments, we conduct further experiments to examine the connections between DN1a and LNd neurons. Our investigation unveiled significant circadian variations in these potential synaptic connections. Notably, our findings indicate that connections between DN1a and LNd neurons are most prominent in the evening, coinciding with the peak of DN1a-LNd inhibitory link in the early evening.

Furthermore, a previous study from the Rosbash lab (Figure 4e from Figure 3b of Guo et al. Nature, 2016) showed diurnal differences in mGluRA expression on LNd, with the highest levels observed in the evening¹. Given that DN1a inhibits LNd via glutamate, the increased inhibition of LNd by DN1a in the evening is consistent with the increased mGluRA expression on LNd during this time.

It's worth mentioning that, due to the unavailability of a specific LexA strain for DN3 labeling, we utilized AstC-LexA to label DN3 which also labels several DN1ps, to conduct the GRASP experiment with DN1a split-GAL4.

Interestingly, we observed that the connection between DN1a and DN3 is stronger at noon and midnight. However, our data demonstrated that DN3 is inhibited by light during the day. Thus, the DN1a-DN3 GRASP data suggest that the DN1a-DN3 inhibitory association may dominate during the night. This part of the data has now been added to the Supplementary Fig. 5a, b in the manuscript.

[REDACTED]

Figure 4 Plastic connections between DN1a, LNd, and DN3

a, c: GRASP signals revealing plastic connections between DN1a and LNds (**a**) or DN3s (**c**).

b, d: Quantification of the GRASP fluorescence between DN1a, LNd (**b**), and DN3 (**d**).

e: Circadian variation in mGluRA mRNA levels in E cells, with a peak at mid-day. Data from

¹.

4) Considering that DN1a's suppress LNd's, can the authors clarify why the calcium levels in both DN1a and LNd's are low during the night?

Thanks to the reviewer's comments, while the calcium signal of DN1a is significantly low at night, the disappearance of light at night leads to an increase in the calcium signal of DN3 itself. In addition, activation of DN3 can inhibit LNd, as shown in the figure below. This may explain the low calcium signal of LNd at night due to DN3 inhibition.

Figure 5 Functional imaging data reveal DN1a inhibit DN3

a Representative pseudocolored images of calcium responses of LNds before and after ATP perfusion activation of DN3s (left). Representative GCaMP traces ($\Delta F/F_0$) of LNds in response to DN3s activation (right, n=12). Scale bars, 10 μ m.

b Quantification of relative max calcium activities of LNds in (a). Data are presented as mean in Max $\Delta F/F_0$ (%) \pm SEM in the histogram; paired t test; **P < 0.01.

Reviewer #2 (Remarks to the Author):

General comments

In this study, the authors discovered that DN1as are most temperature sensitive neurons in central circadian circuit and show diurnal temperature responses. DN1as target two circadian subjects, LNds and DN3s to modulate activity and sleep in different ambient temperatures. Many techniques are employed such as 3D printing, two photon confocal imaging, EM connectome analysis and trans-Tango etc. The designs of research are straightforward. The results are convincing. However, there are some concerns need to be addressed.

Major comments:

1. The researchers posit that DN1as exhibit the highest degree of sensitivity among circadian neurons in response to shifts in ambient temperature. This finding contradicts a previous study conducted by Yadlapalli et al. in 2018[1]. The authors attribute this discrepancy to variations in methodology. However, I note that the authors and Yadlapalli et al. conducted experiments within different time frames. Yadlapalli et al. introduced a temperature elevation to 29 °C in a span of 7 seconds, maintaining it for only 5 seconds upon light activation (as per their methods). Conversely, the authors of this study omitted precise details regarding the duration of their temperature shifting and captured the GCaMP signal at a temporal resolution of minutes.

It is essential to recognize that neurons might exhibit diverse responses to brief temperature fluctuations and to sustained temperature alterations, potentially contributing to the observed disparities, as suggested by Jin et al. in 2021[2]. Therefore, it is imperative for the authors to delineate, during their results discussion, that the conclusions drawn are applicable solely to the specific experimental conditions and temporal scales employed.

Thanks to the reviewer's comments. In response to your suggestions, we have carefully incorporated specific experimental conditions and time scales into our discussion of the experimental results.

We acknowledge your observation regarding the differential response of circadian

neurons to temperature variations over short and extended time periods. The small size of the fruit fly's head makes it highly susceptible to thermal expansion and contraction during temperature fluctuations, potentially causing displacement deviations during imaging. To address this issue, we employed a volumetric calcium imaging approach, ensuring comprehensive capture of neurons along the Z-axis during imaging. While this ensured a broader view, it limited our ability to achieve finer imaging resolution at the second level.

As detailed in our discussion, we indeed observed that DN1as are more sensitive to temperature changes, consistent with findings from Marco Gallio's lab, particularly within brief time intervals ranging from seconds to minutes. However, our results do not exclude the possibility that DN1ps respond to sustained temperature changes over longer time scales. This is supported by Jin et al.'s findings, where prolonged heating at 29°C led to a gradual increase in calcium levels in a subset of DN1ps. We have incorporated this information into the Discussion section.

We are grateful for your constructive feedback, which has significantly improved the robustness and clarity of our study. Your engagement with our research is invaluable to us.

Based on the findings depicted in Fig 1d and Fig 1g, I concur with the evident changes in calcium signaling within DN1a. Nevertheless, the authors have omitted an analysis of the responses exhibited by DN1p and DN2 neurons. The outcomes indicate that distinct DN1p and DN2 neuronal populations might react differentially to temperature stimuli. This discrepancy should not be misconstrued to imply insignificance or insensitivity to thermal cues. Rather, it suggests that the underlying mechanisms are more intricate than the discourse presented by the authors in lines 379-380.

Overall, the manner in which the authors present and discuss their findings warrants a greater degree of prudence and moderation.

Thank you for your comments. We have revised our discussion in line with your suggestions. As you highlighted, a subset of DN1p and DN2 did demonstrate a response to temperature changes in our experiments. However, it is noteworthy that the response in DN1p and DN2 is less pronounced than that in DN1a during short-term temperature changes. Therefore, we did not extensively investigate this particular response. Recognizing the heterogeneity of DN1ps in terms of morphology, connectome, transcriptome, and function (Dingbang Ma, et al. *Elife*. 2021; Xi Jin, et al. *Curr Biol*. 2021; Fang Guo, et al. *Neuron*. 2018), future experiments will require more specific DN1p split-GAL4 lines to identify the temperature-sensitive subset of DN1p and explore the associated input and output circuits. Your insights have been instrumental in refining our work.

2. Indicating the experimental procedures with ZT time notation would be highly beneficial, as the outcomes of the experiments could potentially be influenced by circadian rhythmic factors. For instance, specifying the ZT time at which the experiments illustrated in Figures 1d to 1h were conducted would provide essential contextual information.

Thanks to the reviewer for the comments. As per your request, we have incorporated the ZT time point into the Fig. 1d-k.

3. It is interesting that, whether the LN_d or DN₃, by themselves, can respond to temperature changes, if the synaptic nerve transmission was blocked by tetrodotoxin (TTX) to rule out the effect from the circuit/potential upstream neurons.

Thanks to the thoughtful comments, we conducted an assay to investigate the temperature response of LN_d and DN₃ after applying TTX to block neurotransmitters. The results are depicted in the figure below, revealing no significant change in the calcium signal of LN_d or DN₃ in response to temperature fluctuations. This suggests that the temperature signals perceived by LN_d or DN₃ stem from the input of upstream neurons, including DN_{1a}, and that LN_d or DN₃, in isolation, does not possess the capability to respond to changes in temperature. This portion of the data has now been incorporated

into Supplementary Fig. 5c-d and 9e-f in the manuscript.

Figure 6 Suppression of LNd and DN3 responses to temperature changes by TTX blockage of neuronal transmission

a, c representative GCaMP traces ($\Delta F/F_0$) of LNds (**a**) or DN3s (**c**) in response to temperature changes in the intact fly (blue or red) and add TTX (grey)

b, d Quantification of the relative fold change of calcium activities of LNds in (**a** and **c**).

4.

Regarding Figure 2a, can it be confirmed if these neurons originate from the same brain? If not, how have potential inter-individual variations been accounted for? Concerning Figure 2b, could you clarify if these recordings span 24 hours continuously or occur at six-hour intervals with fly changes? Additionally, it would be helpful to know the number of flies averaged for these recordings.

Thanks to the helpful comments, it appears that our previous description lacked clarity. To provide further clarification, during our analysis, we did not conduct continuous imaging of flies for 12 hours. Instead, we measured DN1a calcium levels in the same flies glued onto imaging chambers every 30 minutes, with each imaging session lasting less than 2 minutes. The flies with chambers were then returned to their original

LD cycle in the incubator. The entire procedure was completed within a 6-hour timeframe.

We have now clearly outlined our long-term imaging protocol in the Methods section:

“3- to 5-day-old *R14F03-GAL4.AD* \cap *VT002963-GAL4.DBD* > *UAS-tdTomato*; *UAS-GCaMP7s* flies were collected under a 12:12 hours LD cycle at 24 °C. The flies were then divided into four groups, each subjected to a 3-day LD entrainment with a 6-hour shift, as detailed below.

On day 4, flies from the four groups, each entrained at different circadian time points, were glued into custom chambers for cuticle surgery to expose the DN1as for two-photon calcium imaging. For flies at ZT12-24, the surgery was performed under red light to avoid disruption of the dark cycle. Flies were habituated for at least 5 min prior to imaging. GCaMP and tdTomato signals in the target neurons were then recorded under two-photon microscopy within 2 minutes. After the rapid measurement, the chambers containing the fixed flies were returned to the original light-dark cycle for further entrainment. Each fly was measured repeatedly at 30-minute intervals over a period of 6 hours. Importantly, flies remained alive after the 6-hour experiments, and we ensured a minimum of 6 flies for data from each circadian time point.

We used ImageJ to calculate GCaMP and tdTomato signals using the same region of interest (ROI). The fluorescence signals from both the GCaMP channel and the tdTomato channel within the identical ROI were summed in Fiji/ImageJ. To normalize the signal, the GCaMP intensity was divided by the tdTomato intensity. The normalized calcium signals from the four groups were then plotted together to create a 24-hour curve.”

An intriguing observation arises when comparing these results with prior research, such as[3].

Liang et al. employed a similar methodology and reported heightened DNI neuron activity during the nighttime, contrasting with the present study's assertion of increased DN1a activity in the daytime. A comprehensive explanation and discussion of these discrepancies are warranted.

We appreciate the insightful comments of the reviewers. It's important to highlight the differences between our study and Paul Taghert's data, which were conducted under constant darkness conditions. Paul Taghert's investigations were aimed at studying the intrinsic rhythms of circadian neuronal activities, whereas our experiments were intentionally designed to be performed under light-dark conditions, in line with our LD behavioral data. In addition, we chose not to record the calcium activity of neurons in the same flies continuously for 24 hours under two-photon microscopy to prevent fluorescence bleaching and to preserve the health of the flies.

It's noteworthy that in Paul Taghert's paper, they didn't differentiate between DN1a and DN1p, referring to them collectively as DN1s. Given the predominance of DN1ps (~15) compared to DN1as (2) per each hemisphere, it is likely that the DN1 signal they measured primarily originates from DN1ps. To address this, we employed the DN1p-specific driver *Clk4.1m-GAL4* to assess the daily calcium fluctuation of DN1p in LD. Importantly, our DN1p daily calcium fluctuation pattern recorded by in-vivo two-photon calcium imaging method is similar to those reported by Paul Taghert, with a peak around late night. This data suggested distinct phases of calcium rhythms between DN1a and DN1p (Figure 7 of the reply).

Figure 7 Calcium oscillation patterns of DN1a and DN1p through long-term *in vivo* calcium

recording in live flies.

Left: Fluorescence signal diagrams of GCaMP and tdTomato in DN1p across 4 circadian time points. **Right:** The averaged calcium fluctuation patterns of DN1a and DN1p at different time points during the day. The data was double-plotted to better visualize the pattern.

From the technical level, for example data in Figure 2, it looks like that, lower F0 baseline values of calcium imaging are correlated with higher $\Delta F/F0$. I am wondering if the authors fix all test group's F0 to a certain value (like 800 in 12bit depth grey recording) by adjusting confocal gain value settings, the differences among Max $\Delta F/F0$ values would be the same conclusion.

Thank you for the reviewer's comments. We have carefully addressed the comments by making modifications and supplements to the methods section. Additionally, we have incorporated the statement 'the F0 setting for all flies remained constant and uniform throughout the imaging session' as suggested

5. In reference to Figures 2e-g, where the author asserts a phase-locked daytime-like response in Per mutant flies (as indicated in lines 143-145), I suggest that a rescue experiment is conducted within the time window of ZT 0-2. This specific time frame aligns with the circadian system's maximal suppression for gating DN1a's responsiveness to temperature fluctuations.

Thanks to the thoughtful comments, we conducted a comprehensive evaluation of the temperature response in *per*⁰ mutants, WT, and DN1a-specific PER rescue flies, specifically focusing on ZT 0-2 and ZT 10-14. Our data, consistent with the original findings presented in Figure 2e-g of the manuscript, reaffirm the absence of significant calcium signal changes in DN1a during temperature fluctuations at these specific time windows for *per*⁰ mutants. Notably, we observed more pronounced changes in the calcium signal of DN1a in all PER rescue strains compared to WT flies, suggesting a potential modulation of DN1a's response to temperature by the circadian clock. These updated findings have been integrated into Supplementary Fig. 2i-k within the manuscript.

Figure 8 Endogenous clock modulates the circadian responses of DN1as to cooling

a, b The representative GCaMP traces ($\Delta F/F_0$) of DN1as in response to cooling in flies at ZT 0-2 (**a**) and ZT 10-14 (**b**)

c Quantification of the relative fold change of calcium activities of DN1as in (**a-b**).

6. In Figures 6, the utilization of *UAS-hid* and optical genetics by the authors to validate DNI functionality is notable. However, these two approaches possess limitations. The application of *UAS-hid* results in the ablation of specific components within the circadian network, potentially triggering unintended consequences such as altering the broader circadian pacemakers and yielding comprehensive outcomes. In the context of utilizing optical genetics for locomotion or sleep testing, it's important to acknowledge that light exposure can introduce confounding effects on behavior, as demonstrated by [4] and other reports.

In light of these considerations, it is recommended to explore alternative genetic tools like *Kir2.1*, *TrpA1* or others to manipulate the neuronal activity, thereby offering a further layer of validation for the observed behavioral outcomes. The author used the TNT to block the DN1as signal, but I'd like to see more data related to genetically manipulate the downstream neurons like LNDs or DN3s.

Thanks to the thoughtful comments, we performed experiments with *Kir2.1*, *TNT*, and *TrpA1* expressed in *Dvpdf* or *DN3-splI*. Since the *Dvpdf* driver also labels pdf+ neurons, we also crossed pdf-GAL4 as a control. Our results confirm the behavioral patterns described in the original article (Figure 5h of the manuscript). Specifically,

inhibition of LNd using *Kir2.1* prevented the E peak advance phenotype typically seen in a low temperature environment. This finding suggests that *Kir2.1* inhibited LNd excitation induced by low temperature, thereby halting the manifestation of E peak advance.

Moreover, inhibition of DN3 using *Kir2.1* or *TNT* at room temperature mirrored the locomotion pattern observed at high temperature, characterized by E peak offset delay and reduced nocturnal sleep. This observation suggests that DN3 inhibition effectively imitates a high-temperature phenotype. Additionally, stimulating *TrpAI*-expressing DN3 at night using high temperature completely suppressed the nocturnal activity increase induced by high temperature. This emphasizes the crucial role of DN3 as a central regulatory hub governing change in nocturnal activity due to temperature fluctuations. These results have been incorporated into Supplementary Fig. 6m-o and 10d-f, j-l in the manuscript

Figure 9 LNd and DN3s modulate the activity induced by temperature variations in flies.

- a**, Inhibition of LNDs results in the suppression of cold-induced E-peak advancement
- b, c** Quantification of E peak locomotion (blue) and E peak advance index (magenta) in (**b**) and PDFs inhibited (**c**).
- e, h, k** Quantification of E peak locomotion (red) and E peak offset delay index (brown) in (**d**, **g**, **j**)
- f, i, l** Quantification of the nocturnal sleep reduction index in panels (**d**, **g** and **j**)

d, g Mimicking hot temperature conditions under normal circumstances through the suppression of DN3s.

j Activation of hot-induced activity changes in flies through the inhibition of DN3s

7. In lines 374-382, given the limited presence of only two DN1a neurons on one side, it becomes evident that DN1a constitutes a minor subset of neurons characterized by the 18H11-GAL4 label. Consequently, ascertaining the precise implications generated by the use of the 18H11-GAL4 driver remains challenging. Referring to Figure 3c, it indicates that DN1p neurons are labeled by the *DNa1-spl>tans-tango*. Hypothetically, utilizing the *DN1p-GAL4>tans-tango* driver might also result in the labeling of DN1a neurons (although this assumption warrants further investigation). It is important to note, however, that the inherent interplay within the DN1 neuronal population demands attention. Functional distinctions between DN1a and DN1p might be more complicated. In light of these considerations, I recommend that the authors exercise caution and refrain from prematurely drawing conclusive insights in line 381-382.

Thanks to the reviewer's suggestion, we have removed the part of the discussion containing lines 381-382.

Minor concern:

1. In Figure 1j, the actual ΔT are varied in 4 to 5 different switching programs, however, the authors only show the temperature switching from 24 to 18 or 22 to 30 Celsius degrees. Better to modify the temperature switching curves.

Thanks to the reviewer's suggestion, we have revised the temperature curve in Fig. 1j.

2. There appear to be several errors or oversights in the citations. In line 45-46, the correct reference should be [5] rather than [2]. Similarly, in line 166-167, the appropriate citation is [6]. Additionally, in line 147-148, it is recommended to include a citation to [7] at minimum. As you are submitting to a comprehensive journal, please ensure to provide appropriate

citations for aspects that might be considered common knowledge to you but could be challenging for other readers to comprehend.

Thanks to the reviewer's suggestion, we have updated the citation accordingly.

3. The authors should provide clarification regarding the rate at which the system achieves specific temperature increases or decreases. Additionally, it is essential to address whether the heating or cooling time remains consistent when employing different "steps," as mentioned in lines 96-97.

In response, we have revised the method, incorporating the information 'This temperature change is achieved in approximately 80 seconds.' Additionally, we have emphasized that 'The time and amplitude of the heating or cooling steps during all experiments are consistent.'

The detail revised as the follows: "The temperature of the Peltier element was measured using a thermistor, and a closed-loop feedback system employing proportional-integral-derivative control was utilized to regulate it. Both the long-term calcium monitor and live imaging experiments employed the same heating and cooling stimuli. The temperature control device (SLD70) was purchased from Sunny Precise Instruments (Shanghai) Co., Ltd. The metal flyplate (<https://github.com/lihailiang7794/metal-flyplate>) (Fig. 1b) was modified by Flyplate5 55. For the live imaging tests, the fly was exposed to the surrounding air. Due to thermal resistance between the metal fly plate and the Peltier element, the actual temperature experienced by the fly slightly deviates from that of the Peltier element (Supplementary Fig. 1a). This temperature change is achieved in approximately 80 seconds. However, during the GCaMP trials, where the fly was submerged in saline, the thermal resistance was significantly reduced, resulting in a closer alignment between the AHL's temperature and that of the Peltier element during the heating and cooling processes (Supplementary Fig. 1a). Additionally, the time and amplitude of the heating or cooling steps during all experiments are consistent."

- [1] S. Yadlapalli, C. Jiang, A. Bahle, P. Reddy, E. Meyhofer, and O. T. Shafer, "Circadian clock neurons constantly monitor environmental temperature to set sleep timing," *Nature*, vol. 555, no. 7694, pp. 98–102, Mar. 2018, doi: 10.1038/nature25740.
- [2] X. Jin, Y. Tian, Z. C. Zhang, P. Gu, C. Liu, and J. Han, "A subset of DN1p neurons integrates thermosensory inputs to promote wakefulness via CNMa signaling," *Current Biology*, vol. 31, no. 10, 2021, doi: 10.1016/j.cub.2021.02.048.
- [3] X. Liang, T. E. Holy, and P. H. Taghert, "Synchronous drosophila circadian pacemakers display nonsynchronous Ca²⁺ rhythms in vivo," *Science (1979)*, vol. 351, no. 6276, 2016, doi: 10.1126/science.aad3997.
- [4] D. D. Au et al., "Drosophila photoreceptor systems converge in arousal neurons and confer light responsive robustness," *Front Neurosci*, vol. 17, 2023, doi: 10.3389/fnins.2023.1160353.
- [5] A. Lamaze, A. Öztürk-Çolak, R. Fischer, N. Peschel, K. Koh, and J. E. C. Jepson, "Regulation of sleep plasticity by a thermo-sensitive circuit in Drosophila," *Sci Rep*, vol. 7, 2017, doi: 10.1038/srep40304.
- [6] S. M. Plaza et al., "neuPrint: An open access tool for EM connectomics," *Front Neuroinform*, vol. 16, 2022, doi: 10.3389/fninf.2022.896292.
- [7] J. L. Price, M. E. Dembinska, M. W. Young, and M. Rosbash, "Suppression of PERIOD protein abundance and circadian cycling by the Drosophila clock mutation timeless," *EMBO Journal*, vol. 14, no. 16, 1995, doi: 10.1002/j.1460-2075.1995.tb00075.x.

Reviewer #3 (Remarks to the Author):

The Guo lab shows an interesting topic about temperature representation in the clock circuits. They employ a fascinating new technique with a two-photon microscope, which allows them to map the circadian clock circuits by analyzing the neural activity to temperature. Although several researches suggest that temperature is represented in the clock circuits, the details are still largely vague. The Guo group found that DN1as are activated/inhibited by warm/cold temperatures. DN1as targets two circuits, E cells (activity) and DN3s (sleep). Overall, it is a very interesting paper, but I have several comments that can help improve the quality of this paper.

Main comments:

1. Discrepancies with the Nature paper (Yadlapalli et al., 2018)

There are several discrepancies from the Nature 2018 paper. The authors briefly explain the reasons. However, they need to show/explain more details of the differences between this study and the Nature 2018 paper and discuss why it is different. For example, please describe each cell's activity against warm/cold temperatures and how this data differs from Nature 2018. It would be great if a summary of the differences is provided. The author mentioned a possible difference related to the reporters they used. I wonder if there are any scientific reasons behind this. It sounds like a non-evidence based.

Lane 95: "Overall, our study provides evidence DN1as can detect rapid changes in ambient temperature,"

Why does the author think DN1as can detect "rapid" changes compared to Nature 2018? Is this based on the experiments? If so, please describe it in more detail.

Thanks to the thoughtful comments. As you pointed out, our original description regarding the reporter difference is not fully evidence-based, so we have revised this

paragraph.

The differences between our study and the one published in Nature (Yadlapalli et al., 2018) are mainly due to differences in experimental methods (two-photon and confocal), in vitro and in vivo anatomical approaches, and physiological state of *Drosophila*.

In our study, we used in vivo two-photon imaging to minimize disturbance to the flies, as opposed to exposing them to intense confocal laser illumination. In addition, we performed fly cuticle surgery and observed fly locomotion on an air-supported ball during imaging to maintain a more physiological state. In contrast, Yadlapalli et al. (2018) "immobilized individual flies on a Peltier element and submerged them in HL3 saline solution," a manipulation that may negatively affect fly health. Furthermore, we utilized metal flyplate attached to Peltier element to control the temperature of saline and ambient air, while Yadlapalli et al. heated or cooled saline around the flies.

Furthermore, our research provides a comprehensive examination of different circadian time points in the response of circadian neurons to temperature changes, whereas previous investigations have primarily focused on the afternoon (ZT6-10), possibly because the confocal laser they used can disturb flies during the dark period (ZT12-24).

In adherence to your recommendations, we undertook supplementary investigations into the reaction of circadian neurons (*Clk856-GAL4 > UAS-GCaMP6s*) to temperature variations at ZT6-10. This involved an analysis of the alterations in calcium signals in each detected cell. Nonetheless, our findings differ from those reported by Yadlapalli et al. (2018). As depicted in Figure 10 of the reply, we observed only a modest response of approximately 3-4 DN1p calcium signals, in reaction to temperature changes.

Given the recognized heterogeneity of DN1ps in morphology, connectome, transcriptome, and functionality (Dingbang Ma, et al. *Elife*. 2021; Xi Jin, et al. *Curr Biol*. 2021; Fang Guo, et al. *Neuron*. 2018), it is plausible that only a specific subset of DN1ps responds to temperature changes. Although we are currently unable to precisely identify this temperature-sensitive subset within the DN1ps, future experiments will require the use of more targeted DN1p split-GAL4 lines. This approach will allow the identification of the temperature-sensitive DN1p subset and facilitate further exploration of its

associated input and output circuitry.

Figure 10 *Drosophila* circadian neurons respond to changes in ambient temperature

a Representative pseudocolor images of calcium responses of *Clk856-GAL4 > UAS-GCaMP6s* flies before and after cooling. white arrow indicates DN1as.

b Heat map of the temporal calcium activities of circadian neurons to cooling (24-18°C). The temperature change was labeled bottom pattern.

c The representative GCaMP traces ($\Delta F/F_0$) of DN1as in response to cooling (blue). c related to a.

Also, please discuss the reasons for the "rapid" temperature response. What is the benefit of the fly's physiology?

Thanks to the reviewer for pointing out that we may have over interpreted our results. First, we recognize that the change in DN1a's response to temperature is not a "rapid" process, because the cooling or heating procedure we used take around 80 sec to achieve aimed temperature. Secondly, our data as well as the report from (Michael H. Alpert, et al. Curr Biol. 2020) 2 both indicate that DN1a senses absolute temperature changes rather than the speed of temperature changes. To avoid any potential misunderstanding, we have removed the mentioned sentence.

2. Sleep data

The authors discuss sleep phenotype using locomotor activity data, but these are not perfectly equal. They should use sleep data instead of locomotor activity data for evaluating the sleep phenotypes.

Thanks to the thoughtful comments, we have revised the method as the follows:

“The nighttime sleep reduction index was employed to quantify heating-induced sleep reduction, estimated as follows: Calculate the difference between the average ZT 12-24 sleep on days with temperatures at 22°C and 30°C. The nighttime sleep reduction index is expressed as (average ZT 12-24 sleep on days with 22°C - average ZT 12-24 sleep on days with 30°C) / average ZT12-24 sleep on days with 22°C. This index was derived by averaging the fly's sleep during the night.

Experimental genotypes and corresponding controls underwent comparison using one-way ANOVA and Tukey's honestly significant difference test. An increase in heat-induced nighttime locomotion is indicated when the nighttime sleep reduction index is greater.”

We also utilized the E peak sleep reduction index to quantify heating/cooling-induced changes in sleep during the evening as the follows: “The E peak sleep reduction index served to quantify heating/cooling-induced sleep reduction during the evening. In cold temperature conditions, it was determined as follows: calculate the difference between the average ZT 6-12 sleep on days with temperatures at 18°C and 24°C. The E peak sleep reduction index is then expressed as (average ZT 6-12 sleep on days with 18°C - average ZT 6-12 sleep on days with 24°C) / average ZT6-12 sleep on days with 24°C.

For hot temperature conditions, the index was estimated by calculating the difference between the average ZT 12-18 sleep on days with temperatures at 18°C and 24°C. The nighttime sleep reduction index is expressed as (average ZT 12-18 sleep on days with 30°C - average ZT 12-18 sleep on days with 22°C) / average ZT12-18 sleep on days with 22°C.

The E peak reduction index was calculated by averaging the fly's sleep in the evening. Experimental genotypes and their respective controls were compared using one-way

ANOVA and Tukey's honestly significant difference test. An increase in heat-induced nighttime locomotion corresponds to a higher E peak sleep reduction index.”

We have also recalculated the E peak sleep index, as detailed below figure. Now we have incorporated these data into (Supplementary Fig. 6h, i, l, 7c, f and 10c, i).

Figure 11: Statistics of flies E peak sleep at various temperature conditions

a-g Statistics of E-peak sleep reduction index in flies under various temperature conditions. a related suppl Fig. 6f. b related suppl Fig. 6g. c related suppl Fig. 6k. d related suppl Fig. 7a. e related suppl Fig. 7e. f related suppl Fig. 10a. g related suppl Fig. 10b.

3. Overstated explanations

I feel there are several overstated explanations. The author's interpretation is beyond what the actual data meant. I did not perfectly agree with some of the author's description/explanation.

Here are some examples:

Line 113: "Our findings revealed a rise in calcium levels during the day and a decline at night, which corresponds to natural daily temperature changes (Fig. 2a, b). These observations suggest that DN1a may have evolved in the fly brain as a circadian thermometer to accurately

reflect daily environmental temperature.”

The condition of this experiment was LD cycles at the constant temp, and DN1a shows the fluctuation of Ca levels. It seems not to be the response to the ambient temp but light-dark cycles. I do not agree the author's statement. Please confirm each explanation/statement is non-biased and non-overstated.

We appreciate your valuable suggestion and concur with your perspective. In response, we have implemented the requisite amendments to our manuscript, specifically in lines 124-127. The pertinent modifications are delineated as follows: 'Our findings unveil a diurnal pattern in the calcium levels of DN1as, characterized by an increase during the day followed by a decrease at night. This rhythmic pattern aligns with the natural light-dark cycle and suggests potential regulation by an intrinsic circadian clock.'

Lane 146: “This data supported our hypothesis that the circadian clock gated the temperature response, as per mutant flies mimic the daytime conditions in which wild-type flies have low PER levels in circadian neurons.”

I understand that the circadian clock gated the temperature response. When PER is expressed more, the neural activity is lower; but why do the authors mention it is related to the PER level? Is there any scientific evidence? It may be just a coincidence.

Thanks to the thoughtful comments, we have revisited our previous findings, with particular attention to the data on the *per⁰* mutant and the rescue of PER (Figure 2 of the manuscript), as shown in Figures a-b below. We think that our data suggest that DN1a activity and PER levels are more than a correlation or coincidence. As in *per⁰* flies, DN1a have higher basal calcium levels both day and night, whereas in DN1a>PER rescue flies, PER levels were overexpressed (Figure 12a-c of the reply), and in this scenario, DN1a basal calcium levels were always lower both day and night. Taken together, manipulation of PER levels in DN1a can dramatically alter the basal circadian activity of DN1a as well as the diurnal calcium response to environmental temperature fluctuations.

To gain a deeper understanding of the mechanism of how PER affects DN1a activity,

we performed a comprehensive screen targeting potential proteins that have been reported to be associated with the circadian clock to regulate circadian neuronal activity or temperature process, including narrow abdomen (Bridget C. Lear, et al. *Neuron*. 2005), shaker (Philip Smith, et al. *J Physiol*. 2019), nocte (Chenghao Chen, et al. *Curr Biol*. 2018), and SERCA (Liang et al. *PNAS*. 2022), as shown in Figure 13 a-d of reply. Notably, when SERCA was specifically knocked down in DN1a, we observed that DN1a lost the circadian variation of calcium levels, with significantly lower basal calcium levels in both day and night. Consequently, these flies showed a bigger response to temperature changes during day. This result is consistent with Liang's data showing that SERCA RNAi in circadian neurons can decrease the amplitudes of calcium fluctuations in circadian neurons.

In addition, our behavioral data showed that specific knockdown of SERCA in DN1a, which reduces DN1a activity, can effectively mimic a cold phenotype, specifically an E peak advance at normal temperature. These flies can further advance the E peak when temperatures are lowered.

Taken together, these data suggest a regulatory role for PER in modulating calcium rhythm in DN1a, as well as the rhythmicity of DN1a in response to temperature fluctuations, possibly through the ER calcium channel SERCA. Considering that we already have 7 main figures and more than 10 supplemental figures, the detailed mechanism will be investigated in the future. This part of the data has now been added to the Supplementary Fig. 3 in the manuscript.

Figure 12 Endogenous clock modulates the circadian responses of DN1as to cooling

a, b PER staining in the brains of wild-type flies (**a**) and DN1a-specific PER-rescued flies (**b**).

c, Quantification of PER level in DN1as.

d, e The representative GCaMP traces ($\Delta F/F_0$) of DN1as in response to cooling in flies at ZT 6-7 (**d**) and ZT 17-19 (**e**)

f, g **f**: Quantification of the relative fold change of calcium activities of DN1as in (**d-e**). **g**: Basal GCaMP6s signals in DN1as at two circadian time points in (**d-e**).

Figure 13 Circadian clock modulate the DN1a diurnal calcium response to ambient temperature variations via SERCA.

a, b The representative GCaMP traces ($\Delta F/F_0$) of DN1as in response to cooling in flies at ZT 6-8 (**a**) and ZT 17-19 (**b**). green curve: wild type; red curve: *per* mutant; purple curve: PER rescue. Dark shades indicate lights off (night); $n \geq 6$.

c, d (**c**): Quantification of the relative fold change of calcium activities of DN1as in (**a-b**). (**d**): Basal GCaMP6s signals in DN1as at two circadian time points in (**a-b**). Data are presented as mean in $\Delta F/F_0$ (%) \pm SEM in the histogram; Tukey's honest significance difference test to assess

the statistical significance of the mean values within each experimental group. In the provided histograms labeled A, B, C and D, the use of the same letter denotes the absence of a significant difference between the two groups, whereas differing letters signify a $p < 0.05$, indicating a significant distinction between the groups. The baseline calcium level represents the spontaneous calcium level of DN1a before temperature fluctuations.

e, Representative locomotor activity profiles of control (n=24) and SERCA knockdown(n=24) flies at 24°C (grey column) and 18°C (blue column). Activity summed in 30-min bins were plotted. Bar graphs are averages \pm SEM; Dark shades indicate lights off (night).

f, Quantification of E peak locomotion and E peak advance index (magenta) in (e). The grey and blue column indicates flies' locomotion at 24°C and 18°C; the purple plots indicate E peak advance; Data are presented as an average locomotion \pm SEM in the histogram; Tukey's honest significance difference test to assess the statistical significance of the mean values within each experimental group. In the provided histograms labeled A, B, C and D, the use of the same letter denotes the absence of a significant difference between the two groups, whereas differing letters signify a $p < 0.05$, indicating a significant distinction between the groups. ** $P < 0.01$.

4. Missing information about the experiments

Line 541: Long-term Intracellular Calcium Measurements

How does in vivo long-term intracellular calcium measurements are measured? Also, it is not clear how to measure it under the light. I can't find the details in the material and method or other places.

Thanks to the helpful comments, it appears that our previous description lacked clarity. To provide further clarification, during our analysis, we did not conduct continuous imaging of flies for 12 hours. Instead, we measured DN1a calcium levels in the same flies glued onto imaging chambers every 30 minutes, with each imaging session lasting less than 2 minutes. The flies with chambers were then returned to their original LD cycle in the incubator. The entire procedure was completed within a 6-hour timeframe. We have now clearly outlined our long-term imaging protocol in the Methods section:

“3- to 5-day-old *R14F03-GAL4.AD* \cap *VT002963-GAL4.DBD* $>$ *UAS-tdTomato*; *UAS-GCaMP7s* flies were collected under a 12:12 hours LD cycle at 24 °C. The flies were then divided into four groups, each subjected to a 3-day LD entrainment with a 6-hour shift, as detailed below.

On day 4, flies from the four groups, each entrained at different circadian time points, were glued into custom chambers for cuticle surgery to expose the DN1as for two-photon calcium imaging. For flies at ZT12-24, the surgery was performed under red light to avoid disruption of the dark cycle. Flies were habituated for at least 5 min prior to imaging. GCaMP and tdTomato signals in the target neurons were then recorded under two-photon microscopy within 2 minutes. After the rapid measurement, the chambers containing the fixed flies were returned to the original light-dark cycle for further entrainment. Each fly was measured repeatedly at 30-minute intervals over a period of 6 hours. Importantly, flies remained alive after the 6-hour experiments, and we ensured a minimum of 6 flies for data from each circadian time point.

We used ImageJ to calculate GCaMP and tdTomato signals using the same region of interest (ROI). The fluorescence signals from both the GCaMP channel and the tdTomato channel within the identical ROI were summed in Fiji/ImageJ. To normalize the signal, the GCaMP intensity was divided by the tdTomato intensity. The normalized calcium signals from the four groups were then plotted together to create a 24-hour curve.”

*Fig. 2 legend, Lane 623: “the results are shown in the histograms labeled A, B, C, D, E and F.”
The statistical analysis is unclear.*

Thank you for your suggestion. We have revised the figure legend as the follows: “In

the provided histograms labeled A, B, C and D, the use of the same letter denotes the absence of a significant difference between the two groups, whereas differing letters signify a $p < 0.05$, indicating a significant distinction between the groups.”

We now incorporated the description into each figure legend.

Lane 645 g, h (g): “Quantification of the relative fold change of calcium activities of DN1as in (e-f). (h): Basal GCaMP6s signals in DN1as at two circadian time points in (e-f). Data are presented as mean in $\Delta F/F_0$ (%) \pm SEM in the histogram; Tukey's honest significant difference tests were used to determine the statistical significance of the means within each experimental group, and the results are shown in the histograms labeled A, B, C and D. Pairwise comparisons were made, and all differences between means were significant at $P < 0.05$.”

The statistical analysis is unclear. What “baseline calcium level” in (h) is?

Thank you for your suggestion. We have revised the figure legend to provide a more accurate description (line 723-730). The baseline calcium level represents the “spontaneous calcium level of DN1a before temperature fluctuations.”

We now revised the figure legend as the follows: “g, h (g): Quantification of the relative fold change of calcium activities of DN1as in (e-f). (h): Basal GCaMP6s signals in DN1as at two circadian time points in (e-f). Data are presented as mean in $\Delta F/F_0$ (%) \pm SEM in the histogram; Tukey's honest significance difference test to assess the statistical significance of the mean values within each experimental group. In the provided histograms labeled A, B, C and D, the use of the same letter denotes the absence of a significant difference between the two groups, whereas differing letters signify a $p < 0.05$, indicating a significant distinction between the groups. The baseline calcium level represents the spontaneous calcium level of DN1a before temperature fluctuations.”

Lane 291 “As expected, light can universally suppress DN3 activity at midday or midnight (Fig. 6f, g).”

The experimental procedure is unclear. How did the authors test the effect of the light in ZT6-8? Fig. 6 figure legend is unclear. h. The explanation is poor and hard to understand, and the

figure legend for (g) is missing.

Thanks to the reviewer's comments, we have adjusted the method as follows: “Male flies aged 3-5 days were maintained for three days under a 12:12 LD (light-dark) cycle at 24°C. On the fourth day, they were glued during a specific time window (ZT 6-8 and ZT17-19) and subjected to two-photon calcium imaging. During the imaging process, a 1 Hz, 0.1 mW/mm² white LED light stimulus was applied. It's important to note that when dissecting flies at ZT17-18, this procedure was conducted under red light. Additionally, it's crucial that the light stimulus doesn't directly illuminate the neuron cell bodies during the light stimulation phase of the experiment.”

Additionally, we have provided a more detailed explanation of the h-graph (line 827-829): “h Representative locomotor activity profiles of control (n=24) and DN3-ablated (n=24) flies at 22°C (grey column) and 30°C (red column). Activity summed in 30-min bins were plotted. Bar graphs are averages ± SEM; Dark shades indicate lights off (night).”

“We also revised figure legend g: “Quantification of the relative fold change of calcium activities of DN3s in (f). Data are presented as mean in $\Delta F/F_0$ (%) ± SEM in the histogram; paired t-test; **P < 0.0001.”**

Minor comments:

Lane 33: 11 and 12 references do not include the feeding behavior.

Thanks to the reviewer's comments, we have revised the original article by removing the phrase "feeding behavior."

Lane 175: “The DN1a-E cell circuit is an intriguing focus of study because the LNds and 5th s-LNv are the primary neurons innervated by DN1as, and these E-cells control the evening locomotion peak, which is dynamically reshaped by temperature.”

There is no citation.

Thanks to the reviewer's comments, we have incorporated the suggested citations (line 188).

Lane 213: “As noted earlier, the calcium levels in LNds exhibit a peak during the evening, and the regulation of the onset and offset of the evening locomotor peak, which is under the control of LNds, can be influenced by temperature variations.”

Which data is talking about?

Thanks to the reviewer's valuable comments, we've accurately refined the original article and incorporated the appropriate literature citations (line 231-234). This sentence now was changed to “Prior research has established that LNd calcium levels display peaks during the night². Temperature changes may potentially affect the regulation of the onset and offset of the nocturnal locomotor peak under LNd control³⁻⁵” according to your suggestions

Lane 273; DN3-like neurons (SLP266, SLP267, and SMP232)

Please explain more. What are they?

Thanks to the reviewer's comments, we have now included the electron microscope connection diagram for SLP266, SLP267, and SMP232 in Supplementary Figure 9a of the manuscript. It's important to mention that SLP266, SLP267, and SMP232, have been derived from the neuPrint nomenclature.

Lane 354: “Notably, we unveiled the pivotal role of DN1a in regulating the activity of the locomotor- and feeding- promoting LNds during the evening, as well as the sleep-promoting DN3s during the night. This exquisite regulation ensures the promotion of locomotion and

feeding during cold periods.”

Where the “feeding information” is coming from?

Thanks to the reviewer's suggestion, we have removed "feeding information" from line 378-380.

Lane 274: “Additionally, the data indicated that DN1as and light input neurons, such as aMe and s-LNv, potentially exert a significant impact on DN3-like neurons, which implies a probable correlation between DN3 activity and its modulation by light and temperature.”

The reference is missing.

Thanks to the reviewer's comments, we have incorporated the suggested citations (line 301). This sentence now was changed to “potentially exert a significant impact on DN3-like neurons, which implies a probable correlation between DN3 activity and its modulation by light and temperature^{4,6,7}.” according to your suggestions.

Lane 285: “Connectome analysis also suggested that DN3s are modulated by light, and we have observed that the reduction in nocturnal sleep induced by heat occurs predominantly after the onset of darkness (ZT12).”

Which data is talking about?

Thanks to the reviewer's suggestion, we have included figure numbers in the original article (line 310-312). This sentence now was changed to “Connectome analysis also suggested that DN3s are modulated by light (Fig. 6a), and we have observed that the reduction in nocturnal sleep induced by heat occurs predominantly after the onset of darkness (ZT12, Supplementary Fig. 10a-c).” according to your suggestions.

- 1 Guo, F. *et al.* Circadian neuron feedback controls the *Drosophila* sleep--activity profile. *Nature* **536**, 292-297, doi:10.1038/nature19097 (2016).
- 2 Liang, X., Holy, T. E. & Taghert, P. H. Circadian pacemaker neurons display cophasic rhythms in basal calcium level and in fast calcium fluctuations. *Proc Natl Acad Sci U S A* **119**, e2109969119, doi:10.1073/pnas.2109969119 (2022).
- 3 Alpert, M. H., Gil, H., Para, A. & Gallio, M. A thermometer circuit for hot temperature adjusts *Drosophila* behavior to persistent heat. *Curr Biol* **32**, 4079-4087.e4074, doi:10.1016/j.cub.2022.07.060 (2022).
- 4 Miyasako, Y., Umezaki, Y. & Tomioka, K. Separate sets of cerebral clock neurons are responsible for light and temperature entrainment of *Drosophila* circadian locomotor rhythms. *J Biol Rhythms* **22**, 115-126, doi:10.1177/0748730407299344 (2007).
- 5 Lorber, C., Leleux, S., Stanewsky, R. & Lamaze, A. Light triggers a network switch between circadian morning and evening oscillators controlling behaviour during daily temperature cycles. *PLoS Genet* **18**, e1010487, doi:10.1371/journal.pgen.1010487 (2022).
- 6 Li, M. T. *et al.* Hub-organized parallel circuits of central circadian pacemaker neurons for visual photoentrainment in *Drosophila*. *Nat Commun* **9**, 4247, doi:10.1038/s41467-018-06506-5 (2018).
- 7 Song, B. J., Sharp, S. J. & Rogulja, D. Daily rewiring of a neural circuit generates a predictive model of environmental light. *Sci Adv* **7**, doi:10.1126/sciadv.abe4284 (2021).

REVIEWER COMMENTS

Reviewer #1 (Remarks to the Author):

The authors performed additional experiments in response to my comments, but the results ended up sometimes contradictory to their previous claims. The questions about underlying mechanisms have not been answered adequately.

Major comments:

- In my comment #2, I asked the authors to provide more details about how DN1a's temperature response might be influenced by the circadian clock. The additional experiments they did provided more confusing results.

The authors initially reported that in *per01* flies, DN1a's have higher basal calcium levels both day and night, whereas when PER levels were overexpressed DN1a basal calcium levels were always lower both day and night.

In Liang 2022 PNAS paper, the authors showed that PER protein rhythms are diminished and PER protein levels are always low when SERCA is knocked down in all pacemaker neurons, similar to *per01* condition. However the authors report that "when SERCA was specifically knocked down in DN1a, we observed that DN1a lost the circadian variation of calcium levels, with significantly lower basal calcium levels in both day and night.", which is inconsistent with their earlier results. I would have expected SERCA knockdown flies to have similar results as *per01* flies, which is higher basal calcium levels throughout the circadian cycle. However, they observed opposite phenotypes in both these conditions.

Their additional experiments led to contradictory results and do not provide mechanistic understanding of how PER levels impact basal calcium rhythms.

- In my comment #3, I asked the authors to explain how the inhibitory link between DN1a and LNd might be stronger in the early evening, whereas the DN1a-DN3 inhibitory connection might dominate during the night.

Their additional GRASP experiments in wildtype flies further validated their claim, but did not provide any explanation for 'how' this could be happening. They talk about mGluRA expression in LNd, but did not provide any functional data. The authors did not address the original mechanism question.

- In my comment #4, "Given the author's claim that DN1a's suppress LNd's, can they clarify why the calcium levels in both DN1a and LNd's are low during the night?"

They show that activation of DN3 might inhibit LNd, and could be the possible explanation for why LNd calcium levels are low at night.

I think their original claim that DN1a's suppress both the LNds that promote activity and the DN3s that promote sleep seems too simplistic and doesn't take into account all the potential connections between other clock neurons.

Reviewer #2 (Remarks to the Author):

The authors have addressed all my questions. The presentation of this manuscript has been largely improved. I have no further questions.

Reviewer #3 (Remarks to the Author):

The authors answered all my questions. The manuscript was also much improved. Thank you that the authors put a lot of effort on it. However, just my last comment is that I wonder if the authors could include the Figure 10 in the rebuttal letter in the supplemental figures. I may have missed it, but I could not find the descriptions regarding this comment nor Figure 10 in the manuscript. Figure 1 itself gives a negative impression of Shafer's result, since DN1ps show no response to temperature. However, the new data (Fig. 10) nicely shows that DN1ps show a response at ZT 6-8, suggesting that DN1ps has a different response to temperature depending on the time of day. Importantly, the Gao lab reproduces Shafer's data. The data support both Shafer's and Gao's findings, which are very important observations. Therefore, I suggest that the author include the data in the Supplementary Figures. People generally agree that DN1ps are a mixed type of neuronal population.

Reviewer #1 (Remarks to the Author):

In my comment #2, I asked the authors to provide more details about how DN1a's temperature response might be influenced by the circadian clock. The additional experiments they did provided more confusing results.

*The authors initially reported that in *per01* flies, DN1a's have higher basal calcium levels both day and night, whereas when *PER* levels were overexpressed DN1a basal calcium levels were always lower both day and night.*

*In Liang 2022 PNAS paper, the authors showed that *PER* protein rhythms are diminished and *PER* protein levels are always low when *SERCA* is knocked down in all pacemaker neurons, similar to *per01* condition. However the authors report that "when *SERCA* was specifically knocked down in DN1a, we observed that DN1a lost the circadian variation of calcium levels, with significantly lower basal calcium levels in both day and night.", which is inconsistent with their earlier results. I would have expected *SERCA* knockdown flies to have similar results as *per01* flies, which is higher basal calcium levels throughout the circadian cycle. However, they observed opposite phenotypes in both these conditions.*

*Their additional experiments led to contradictory results and do not provide mechanistic understanding of how *PER* levels impact basal calcium rhythms.*

Thank you for your comments. We recognize the importance of clarifying the potential inconsistency regarding the similarity between the results of knocking down *SERCA* in DN1a and those observed in *per0* flies.

Upon a thorough reexamination of Liang et al.'s data and after engaging in direct communication with Liang, we first aim to address Reviewer 1's conclusion that "PER protein levels are always low when *SERCA* is knocked down in all pacemaker neurons."

In Figure 1, we merged the right panels from Figure 4A and 4B in Liang's paper to facilitate a more comprehensive comparison of *PER* levels in DN1s, which include our primary interest, DN1as. The examination reveals that pan-circadian neuronal

knockdown of SERCA diminishes PER rhythms and PER levels in LN_v and LN_d. Importantly, in both DN1 and DN3, *tim-GAL4>SERCA RNAi* flies demonstrate a reduction in PER protein rhythms, primarily attributed to a significantly low PER level at ZT0 (Figure 1a-b of the reply). However, at ZT12, the PER levels in DN1 and DN3 of *tim-GAL4>SERCA RNAi* flies surpass those of the control, as highlighted by the arrow (Figure 1c of the reply). These findings suggest that pan-circadian neuronal knockdown of SERCA results in a reduction in PER cycle amplitude while maintaining a relatively steady PER level in DNs, including DN1as. This pattern notably differs from *per0* flies, where PER levels are undetectable at all circadian times.

[REDACTED]

Figure 1 PER protein rhythms of control flies and flies with SERCA knocked down in all pacemaker neurons.

a, Averaged PER protein staining intensity at four different time points (ZT0, ZT6, ZT12, and ZT18) in DN1s and DN3s from control flies (this data from Xitong Liang, et al. 2022. PNAS).

b, PER protein rhythms are diminished when knocking down SERCA in all pacemaker neurons by *tim-GAL4* (KK107371) (this data from Xitong Liang, et al. 2022. PNAS).

c, Merge of Figure 1a and b.

Additionally, caution should be exercised when extrapolating conclusions from *tim-GAL4 > SERCA RNAi* flies to other *GAL4 > SERCA RNAi* flies, given the variability in GAL4 driver line strength and expression patterns. In Liang's study, SERCA knockdown using *tim-GAL4* resulted in high rates of lethality (69% in *tim > KK10737 SERCA RNAi*) and arrhythmicity (50% in *tim > JF01948 SERCA RNAi*, and 73% in *tim > KK10737*

SERCA RNAi), as reported in Table 1 of Liang et al.'s paper. However, *SERCA* knockdown using *pdf-GAL4* had little effect on arrhythmicity (0% in *pdf > JF01948* and 13% in *pdf > KK107371*, data from Table 1 of Liang et al.'s paper) compared to a positive hit (65% arrhythmicity in *pdf > KK100082 α1T RNAi*, data from Table 1 of Liang et al.'s paper).

If *SERCA* is knocked down by *pdf-GAL4*, resulting in diminished *PER* protein rhythms and consistently reduced *PER* protein levels in *LN_v*—resembling the observations in *LN_v* from *tim-GAL4 > SERCA RNAi* flies—it is anticipated that *pdf-GAL4 > SERCA RNAi* flies would display some level of arrhythmicity, akin to the arrhythmic phenotype observed in *per01* only in the *PDF*-expressing cells (Figure 4c of Dan Stoleru, et al. 2004. Nature). Therefore, the statement regarding “*PER* protein rhythms are diminished and *PER* protein levels are always low when *SERCA* is knocked down in all pacemaker neurons by *tim-GAL4*” may not be applicable to other *GAL4* drivers, such as *Pdf-GAL4* or *R23E05-GAL4*.

Figure 2 knockdown of *SERCA* leads to a reduction in the oscillation of the circadian protein *PER*.

a *PER* staining in the brains of *R23E05 > Dicer2, UAS-GCaMP6s* (left) and *R23E05 > Dicer2, UAS-GCaMP6s + SERCA RNAi* flies (right) at ZT11 and ZT 23. Scale bars, 20 μ m. The top panel indicates the time of staining.

b Quantification of *PER* level in *DN1as*, *LNd* and *LN_v* (a, $n \geq 6$). Data are presented as average fluorescence of *PER* \pm SEM in the histogram; Tukey's honest significance difference test to assess the statistical significance of the mean values within each experimental group. In the

provided histograms labeled A, B, C, D and E, the use of the same letter denotes the absence of a significant difference between the two groups, whereas differing letters signify a $p < 0.05$, indicating a significant distinction between the groups.

To further investigate and validate our hypothesis, we employed *R23E05-GAL4* to selectively express SERCA RNAi in DN1a, examining PER protein rhythm and levels across all circadian neurons at ZT11 and ZT23. Our analysis included the evaluation of PER expression in DN1a, LNd, and LNv. The presented figure illustrates that when SERCA RNAi is expressed in DN1a, PER in DN1a continues to display robust circadian cycling, characterized by higher levels at ZT23 and lower levels at ZT11 (Figure 2 of the reply). This pattern is distinctly different from the '*per01* condition' proposed by reviewer1. The amplitude of PER oscillations in DN1a decreases upon SERCA knockdown (Figure 2b of the reply), with lower levels at ZT23 and higher levels at ZT11 compared to controls, partially mirroring Liang's findings (Figure 1c of the reply). Importantly, targeted expressing SERCA RNAi in DN1a does not induce significant changes in PER protein rhythms in LNd and LNv.

Integrating these findings with our calcium imaging results, we observed that utilizing *R23E05-GAL4* to suppress SERCA in DN1as somewhat sustains PER protein rhythms in DN1a while significantly reducing DN1a basal calcium levels and enhancing the cold-induced response during the day (Supplementary Fig. 3 of the manuscript).

Finally, Reviewer 1 suggested that "SERCA knockdown flies should exhibit similar results to *per⁰¹* flies, characterized by higher basal calcium levels throughout the circadian cycle". However, our model posited a direct pathway from PER to SERCA to basal calcium levels. In other words, SERCA could more directly influence basal calcium levels without necessarily acting through PER. As the DN1a SERCA RNAi flies displayed a phenotype distinct from *per01* flies, we aimed to elucidate the epistasis relationship between SERCA and PER. To address this, we designed a new experiment placing DN1a SERCA RNAi flies in constant light (LL) conditions to continually degrade PER and specifically examine the impact of SERCA on calcium levels and cold response of DN1a.

In this scenario, if SERCA acts through PER to modulate DN1a calcium levels, the DN1a SERCA RNAi flies in LL should exhibit a *per⁰¹*-like phenotype, characterized by high basal calcium levels and weak cold-induced calcium reduction. On the other hand, if SERCA is epistatic to PER on basal calcium regulation, the DN1a SERCA RNAi flies in LL should still exhibit a SERCA knockdown-like phenotype with consistently lower basal calcium levels and enhanced cold-induced calcium reduction throughout the circadian cycle.

Figure 3 Circadian proteins modulate calcium level oscillations in circadian neurons through SERCA

a PER staining in the brains of *R23E05 > Dicer2, UAS-GCaMP6s* (left) and *R23E05 > Dicer2, UAS-GCaMP6s + SERCA RNAi* flies (right) at ZT6 and ZT 18 under LL conditions. Scale bars, 20 μ m. The top panel indicates the time of staining.

b Quantification of PER level in DN1as (a, $n \geq 6$). Data are presented as average fluorescence of PER \pm SEM in the histogram; Tukey's honest significance difference test to assess the statistical significance of the mean values within each experimental group. ns, no significance.

c,d The representative GCaMP traces ($\Delta F/F_0$) of DN1as in response to cooling in flies at ZT 6 (a) and ZT 18 (b). green curve: wild type; red curve: *per* mutant; purple curve: PER rescue. Dark shades indicate lights off (night); $n \geq 6$. The top panel indicates the time of staining and imaging. Right panel: Quantification of the relative fold change of calcium activities of DN1as in left panel. paired t-test; **** $p < 0.0001$.

e Basal GCaMP6s signals in DN1as at two circadian time points in (c-d). Data are presented as mean in $\Delta F/F_0$ (%) \pm SEM in the histogram; Tukey's honest significance difference test to assess the statistical significance of the mean values within each experimental group. In the provided histograms labeled A and B, the use of the same letter denotes the absence of a significant difference between the two groups, whereas differing letters signify a $p < 0.05$, indicating a significant distinction between the groups. The baseline calcium level represents the spontaneous calcium level of DN1a before temperature fluctuations.

f A model illustrating circadian proteins' influence on calcium levels in circadian neurons through SERCA.

Our data clearly demonstrate that during LL, PER were maintained at an undetected level in the fly brain (Figure 3a-b of the reply). The results, as illustrated in the figure below, reveal a noteworthy finding. Under LL conditions, the control group exhibited performance comparable to *per0*. The DN1as in controls displayed a constantly high basal calcium level at both ZT6 and ZT18, accompanied by a relatively feeble response to cold. Conversely, in DN1a SERCA RNAi flies, a distinct pattern emerged. Despite constant low PER level in these flies under LL, the basal calcium level of DN1a remained not only consistently low but also displayed an enhanced cold-induced inhibition (Figure 3c-d in the reply). This observation clearly supports the notion that SERCA serves as a crucial

molecular effector influencing DN1a calcium levels, and the circadian clock may act through SERCA to regulate diurnal calcium changes and cold-induced response.

Importantly, the data also highlight a reciprocal relationship between SERCA and the core circadian protein oscillation (Figure 3f in the reply). This is substantiated by data demonstrating that a robust knockdown of SERCA levels blocks PER oscillation in the *tim-gal4> KK10737* (stronger RNAi line) flies (Figure 1c in the reply). It is crucial to note that employing a less potent GAL4 line, *R23E05-GAL4*, to express KK10737 RNAi in DN1as somewhat sustains PER protein rhythms in DN1a. Furthermore, the use of *R23E05-GAL4* to express either stronger or weaker RNAi lines produces notable and comparable changes in basal calcium levels and cold-induced calcium response (Figure S3 in the manuscript). This suggests that the basal calcium level is highly sensitive to alterations in SERCA levels.

We express our sincere gratitude to the reviewers for their insightful comments, which have greatly contributed to enhancing our understanding of the intricate mechanisms governing the regulation of calcium levels in circadian neurons by circadian proteins, particularly through the role of SERCA.

In response to the reviewers' feedback, we have reorganized the existing figures illustrating SERCA's impact on circadian proteins and calcium levels in circadian neurons. These figures have been incorporated into Supplementary Fig. 4 within the manuscript.

It is crucial to note the substantial volume of data presented in this manuscript, comprising 7 main figures and 12 supplementary figures. The primary focus of our work lies in unraveling the intricate interplay between *Drosophila* circadian neuronal circuit, temperature integration, and the subsequent reshaping of neural and behavioral mechanisms governing sleep-wakefulness. Continuously adding data on mechanisms runs the risk of burdening the reader and diluting the central theme of the article. At this juncture, our supplementary data have only scratched the surface of the underlying mechanism. In our forthcoming paper, we plan to collaborate with the Rosbash lab to conduct a detailed analysis of how the circadian clock, and the ER calcium fluxes influences basal calcium rhythms in circadian neurons. This collaborative effort aims to

provide a more comprehensive elucidation of this regulatory pathway through the utilization of single-cell sequencing data, RNAi screening, and calcium imaging techs.

In my comment #3, I asked the authors to explain how the inhibitory link between DN1a and LNd might be stronger in the early evening, whereas the DN1a-DN3 inhibitory connection might dominate during the night.

Their additional GRASP experiments in wildtype flies further validated their claim, but did not provide any explanation for 'how' this could be happening. They talk about mGluRA expression in LNd, but did not provide any functional data. The authors did not address the original mechanism question.

We appreciate the valuable feedback from the reviewers. In response, we conducted a detailed analysis to address the question of why the inhibitory link between DN1a and LNd might be stronger in the early evening. Firstly, by examining the specific synaptic distribution of DN1a to principal downstream E cells using Neuprint database, we observed that the DN1a to E cell synapses are primarily located in the ventromedial (VM) and ventrolateral (VL) region (Figure 4a-b in the reply). Monitoring the dynamic changes in the GRASP signals between DN1a and E cells at different circadian time points revealed stable signals in the VM region. However, the GRASP signal in the synapses densely distributed VL region exhibited significant dynamic day-night variations, with signals nearly undetectable during the day and a noticeable increase at ZT11, the onset of the evening peak (Figure 4c-d in the reply). Considering the specific synapses from DN1a to E cells in the VL region, this indicates a specific increase in the synaptic density during this time and explains why the inhibitory link between DN1a and LNd might be stronger in the early evening.

[REDACTED]

Figure 4 Plastic connections between DN1a and LNd

a DN1a synaptic junction projecting to LNd. This data from Hemibrain connectome dataset (neuPrint). a: anterior. l: latter. v: ventrolateral.

b A Sankey histogram depicting the synaptic connections between DN1as and E cells, as determined by the Hemibrain connectome dataset.

c GRASP signaling reveals plastic connections between DN1a and LNd, indicative of pronounced precept plasticity. The cyan signals of VL area indicate the presence of GRASP signaling between DN1as and LNds. VL: ventrolateral; VM: ventromedial. Scale bars, 20 μm

d Quantification of the GRASP fluorescence between DN1a and LNd. Data are presented as mean in $\Delta F/F_0$ (%) \pm SEM in the histogram; Tukey's honest significance difference test to assess the statistical significance of the mean values within each experimental group. In the

provided histograms labeled A, B and C, the use of the same letter denotes the absence of a significant difference between the two groups, whereas differing letters signify a $p < 0.05$, indicating a significant distinction between the groups. $**P < 0.01$.

e Antiphase oscillations in LNV and DN1a axon morphology, measured in light-dark cycles [2 p.m. (ZT6), daytime; 2 a.m. (ZT18), nighttime]. Each dot, one brain hemisphere. (A to E) t test in LNVs, two-way ANOVA with Tukey's post hoc test in DN1as unless stated otherwise. DN1a ventrolateral (VL) tracts change; ventromedial (VM) tracts are internal controls. This data from (Bryan J. Song et al. in Sci Adv 2021)

f Manipulating Rho1 in DN1as bidirectionally changes axonal fasciculation. One-way ANOVA with Tukey's post hoc test. DN1a ventrolateral (VL) tracts change; ventromedial (VM) tracts are internal controls. This data from (Bryan J. Song et al. in Sci Adv 2021).

Regarding the underlying molecular mechanism, we acknowledge the relevant findings reported by Bryan J. Song et al. in Sci Adv 2021. Their work delves into the detailed mechanism of how the circadian clock establishes the rhythmicity of axonal remodeling in DN1a. They observed daily remodeling of DN1a axons, predominantly occurring in the VL region. In the daytime, DN1a fibers in the VL region are shorter, while at night, DN1a axons in the VL region extend (Figure 4e in the reply), aligning with our GRASP results (Figure 4c-d in the reply). The rhythmicity of DN1a axonal remodeling is under the control of the circadian clock, as evidenced by its disappearance in the *per0* mutant. The circadian clock regulates axonal growth through the classical Rho signaling pathway, with Rho overexpression leading to DN1a axon contraction, and RNAi-mediated downregulation of Rho resulting in DN1a axon extension (Figure 4f in the reply). A similar mechanism has been already detailed in Afroditi Petsakou's Cell paper (Afroditi Petsakou, et al. 2015. Cell). Therefore, based on these data, we propose that the circadian clock regulates Rho1 activity in DN1a, promoting axon extension in the VL region during the early evening and leading to stronger inhibitory synapses from DN1a to E cells.

The recently analyzed data are now presented in Supplementary Figure S6. Once again, considering the current volume and the central theme of our manuscript, we believe that continually adding more data on mechanisms could become an endless endeavor. Our

intention is to explore the detailed mechanism in our next paper.

• *In my comment #4, “Given the author’s claim that DN1a’s suppress LNd’s, can they clarify why the calcium levels in both DN1a and LNd’s are low during the night?”*

They show that activation of DN3 might inhibit LNd, and could be the possible explanation for why LNd calcium levels are low at night.

I think their original claim that DN1a’s suppress both the LNds that promote activity and the DN3s that promote sleep seems too simplistic and doesn’t take into account all the potential connections between other clock neurons.

Thank you for your comments. We acknowledge that the lower nocturnal calcium levels observed in DN1a and LNd may be influenced by other potential circadian neurons, including DN1p. In our previous study (Figure 5a of the reply; Fang Guo, et al. 2016, Nature), functional imaging results demonstrated that DN1p activation significantly inhibits calcium levels in LNd. Additionally, both our earlier data (Figure 5b) and findings reported by Xitong Liang et al. (2022, PNAS) (Figure 5c) suggest that DN1p exhibits elevated calcium levels during the night.

This information underscores the complex interplay among various circadian neurons and their dynamic activities throughout the circadian cycle. The potential impact of DN1p activation on LNd calcium levels, coupled with the nocturnal elevation of DN1p calcium, provides additional insights into the intricate regulatory mechanisms governing calcium dynamics in different neuronal populations. In the discussion, we have adjusted our tone and acknowledge the potential complex interplay among various circadian neurons contributing to the daily calcium oscillation pattern of LNds and DN3s.

[REDACTED]

Figure 5 DN1s directly contact and reduce calcium levels in core pacemakers

a, DN1s inhibit calcium levels in the core pacemakers. This data from (Fang Guo, et al. 2016. Nature).

b, The averaged calcium fluctuation patterns of DN1a and DN1p at different time points during the day. The data was double-plotted to better visualize the pattern.

c, raw calcium fluorescence intensity traces from DN1p fly are shown. This data from (Xitong Liang et al. 2022, PNAS)

Reviewer #3 (Remarks to the Author):

The authors answered all my questions. The manuscript was also much improved. Thank you that the authors put a lot of effort on it. However, just my last comment is that I wonder if the authors could include the Figure 10 in the rebuttal letter in the supplemental figures. I may have missed it, but I could not find the descriptions regarding this comment nor Figure 10 in the manuscript. Figure 1 itself gives a negative impression of Shafer's result, since DN1ps show no response to temperature. However, the new data (Fig. 10) nicely shows that DN1ps show a response at ZT 6-8, suggesting that DN1ps has a different response to temperature depending on the time of day. Importantly, the Gao lab reproduces Shafer's data. The data support both Shafer's and Gao's findings, which are very important observations. Therefore, I suggest that the author include the data in the Supplementary Figures. People generally agree that DN1ps are a mixed type of neuronal population.

Thank you for your insightful comments. We have incorporated Figure 10 into Supplementary Figure 1j-k as suggested.

REVIEWERS' COMMENTS

Reviewer #1 (Remarks to the Author):

The authors addressed my questions.